# ControlFusion: A Controllable Image Fusion Network with Language-Vision Degradation Prompts

**Linfeng Tang[1,†],   Yeda Wang[1,†],   Zhanchuan Cai[2],   Junjun Jiang[3],   Jiayi Ma[1,*]**

[1]Electronic Information School, Wuhan University
[2]School of Computer Science and Engineering, Macau University of Science and Technology
[3]Faculty of Computing, Harbin Institute of Technology
linfeng0419@gmail.com,   wangyeda@whu.edu.cn,
zccai@must.edu.mo,   jiangjunjun@hit.edu.cn,   jyma2010@gmail.com,

## Abstract

Current image fusion methods struggle with real-world composite degradations and lack the flexibility to accommodate user-specific needs. To address this, we propose ControlFusion, a controllable fusion network guided by language-vision prompts that adaptively mitigates composite degradations. On the one hand, we construct a degraded imaging model based on physical mechanisms, such as the Retinex theory and atmospheric scattering principle, to simulate composite degradations and provide a data foundation for addressing realistic degradations. On the other hand, we devise a prompt-modulated restoration and fusion network that dynamically enhances features according to degradation prompts, enabling adaptability to varying degradation levels. To support user-specific preferences in visual quality, a text encoder is incorporated to embed user-defined degradation types and levels as degradation prompts. Moreover, a spatial-frequency collaborative visual adapter is designed to autonomously perceive degradations from source images, thereby reducing complete reliance on user instructions. Extensive experiments demonstrate that ControlFusion outperforms SOTA fusion methods in fusion quality and degradation handling, particularly under real-world and compound degradations. The source code is publicly available at `https://github.com/Linfeng-Tang/ControlFusion`.

## 1   Introduction

Image fusion is a crucial technique in image processing. By effectively leveraging complementary information from multiple sources, the limitations associated with information collection by single-modal sensors can be significantly mitigated [41]. Among the research areas in this field, infrared-visible image fusion (IVIF) has garnered considerable attention. IVIF integrates essential thermal information from infrared (IR) images with the intricate texture details from visible (VI) images, offering a richer and more complete depiction of the scene [46]. By harmonizing diverse information and producing visually striking results, IVIF has found extensive applications in areas such as security surveillance [47], military detection [22], scene understanding [44], and assisted driving [1], *etc*.

Recently, IVIF has emerged as a focal point of research, leading to rapid advancements in related methods. Based on the adopted network architectures, these methods can be categorized into convolutional neural network-based [29, 51], autoencoder-based [10, 11], generative adversarial network-based [19, 15], Transformer-based [20, 45], and diffusion model-based [50, 32] methods. In addition, from a functional perspective, they can be grouped into visual-oriented [29, 51], joint registration-fusion [27, 37], semantic-driven [28, 16], and degradation-robust [31, 42, 43] schemes.

---

† Equal contribution;    * Corresponding author.

39th Conference on Neural Information Processing Systems (NeurIPS 2025).

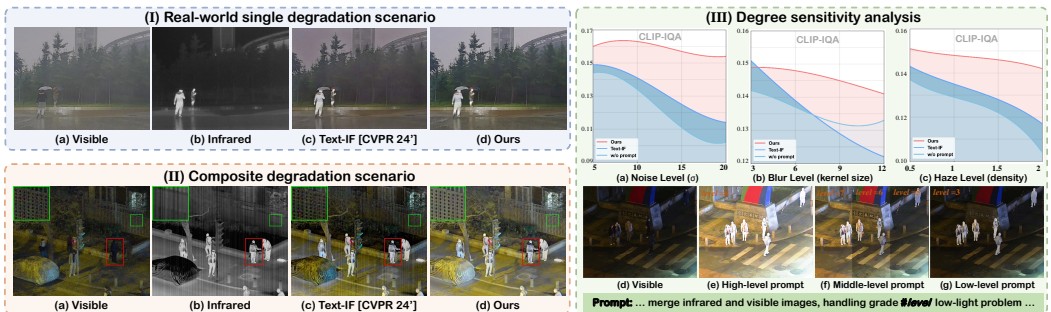

Figure 1: Comparison across real-world, composite degradation, and varying degradation levels.

Notably, although degradation-robust schemes can mitigate degradation to some extent, certain limitations exist. First, existing restoration-fusion methods often adopt simplistic strategies for training data construction, overlooking the domain gap between simulated data and realistic images, which hampers their generalizability in practical scenarios, as illustrated in Fig. 1 (I). Second, they are tailored for specific or single types of degradation, making them ineffective in handling more complex composite degradations, as illustrated in Fig. 1 (II). Finally, as shown in Fig. 1 (III), existing methods lack degradation level modeling, causing a sharp decline in performance as degradation intensifies. Moreover, they lack flexibility to adapt fusion results to diverse user preferences.

To overcome these limitations, we propose a versatile and controllable fusion model based on language-vision prompts, termed ControlFusion. On the one hand, we introduced a physics-driven degradation imaging model that differs from existing approaches by simultaneously simulating degradation processes for both visible and infrared modalities with high precision. This model not only effectively narrows the gap between synthetic data and real-world images but also provides crucial data support for addressing complex multi-modal composite degradation challenges. On the other hand, we develop a prompt-modulated image restoration and fusion network to generate high-quality fusion results. The prompt-modulated module enables our network to dynamically adjust feature distribution based on degradation characteristics (which can be specified by users), achieving robust feature enhancement. This also allows our method to respond to diverse user requirements. Furthermore, we devise a spatial-frequency visual adapter that combines frequency-domain degradation priors to directly extract degradation cues from degraded inputs, eliminating the heavy reliance on user instructions, as customizing prompts for each scene is time-consuming and labor-intensive. As shown in Fig. 1, benefiting from the aforementioned designs, our method excels in handling both real-world and composite degradation scenarios and can flexibly respond to user needs. In summary, our main contributions are as follows:

• We propose ControlFusion, a versatile image restoration and fusion framework that uniformly models diverse degradation types and degrees, using textual and visual prompts as a medium. Specifically, its controllability enables it to respond to user-specific customization needs.
• A spatial-frequency visual adapter is devised to integrate frequency characteristics and directly extract text-aligned degradation prompts from visual images, enabling automated deployment.
• A physics-driven imaging model is developed, integrating physical mechanisms such as the Retinex theory and atmospheric scattering principle to bridge the gap between synthetic data and real-world images, while taking into account the degradation simulation of infrared-visible dual modalities.

## 2 Related Work

**Image Fusion.** Earlier visual-oriented fusion approaches primarily concentrated on merging complementary information from multiple modalities and improving visual fidelity. The mainstream network architectures primarily include CNN-based [14, 48], AE-based [11, 49], GAN-based [19, 15], Transformers [20] and diffusion models [50]. Furthermore, several schemes including joint registration and fusion [33, 37], semantic-driven [28, 15], and degradation-robust [39, 42, 38] methods, are proposed to broaden the practical applications of image fusion. For instance, Tang et al. [30] and Liu et al. [17] developed corresponding solutions for illumination distortions and noise interference, respectively. In addition, [39] proposed a image restoration and fusion network to address various degradations.

However, it fails to handle mixed degradations and struggles to generalize to real-world scenarios. In particular, it relies on tedious manual efforts to customize text prompts for each scene, hindering large-scale automated deployment.

**Image Restoration with Vision-Language Models.** With the advancement of deep learning toward multimodal integration, the image restoration field has progressively evolved into a new paradigm of text-driven schemes. CLIP [25] establishes visual-textual semantic representation alignment through dual-stream Transformer encoders pre-trained on 400 million image-text pairs. This framework lays the foundation for the widespread adoption of *Prompt Engineering* in computer vision. Recent studies integrate CLIP encoders with textual user instructions, proposing generalized restoration frameworks including PromptIR [23], AutoDIR [8], and InstructIR [2], which effectively handle diverse degradation types. Moreover, SPIRE [24] further introduces fine-grained textual restoration cues by quantifying degradation severity levels and supplementing semantic information for precision restoration. To eliminate reliance on manual guidance, Luo et al. [18] developed DA-CLIP by fine-tuning CLIP on mixed-degradation datasets, enabling degradation-aware embeddings to facilitate distortion handling. However, existing unified restoration methods are primarily designed for natural images, which exhibit notable limitations when processing multimodal images (*e.g.*, infrared-visible).

## 3   Physics-driven Degraded Imaging Model

Due to the differences in the imaging mechanisms of infrared and visible, the types of degradation they face also vary. To tackle the complex and variable degradation challenges, we propose a physics-driven imaging model for infrared and visible images aimed at reducing the domain gap between simulated data and real-world imagery. Infrared (IR) images usually suffer from sensor-related interference, such as stripe noise and low contrast. While Visible (VI) images are typically degraded by illumination conditions (low light, over-exposure), weather (rain, haze), and sensor-related issues (noise, blur). Given a clear image $I_m(m \in \{ir, vi\})$, the proposed imaging model can be mathematically represented as:

$$D_m = \mathcal{P}_s \left( \mathcal{P}_w \left( \mathcal{P}_i \left( I_m \right) \right) \right), \tag{1}$$

where $D_m$ is the corresponding degraded image, and $\mathcal{P}_i$, $\mathcal{P}_w$, and $\mathcal{P}_s$ represent illumination-, weather-, and sensor-related distortions, respectively.

**Sensor-related Distortions.** Sensor-related distortions encompass various types of noise, motion blur, and contrast degradation. Contrast degradation and stripe noise of infrared images can be modeled as:

$$D_{ir}^s = \mathcal{P}_s(I_{ir}) = \alpha \cdot I_{ir} + \mathbf{1}_H \mathbf{n}^\top, \tag{2}$$

where $\alpha$ is a constant less than 1 for contrast reduction, $\mathbf{1}_H \in \mathbb{R}^H$ is an all-ones column vector and $\mathbf{n} \in \mathbb{R}^W$ represents the column-wise Gaussian noise vector sampled from $\mathcal{N}(0, \epsilon^2)$ with $\epsilon \in [1, 15]$. Moreover, Gaussian noise and motion blur in source images can be modeled as:

$$D_m^s = \mathcal{P}_s(I_m) = I_m * K(N, \theta) + \mathcal{N}(0, \sigma^2), \tag{3}$$

where $\mathcal{N}(0, \sigma^2)$ represents Gaussian noise with $\sigma \in [5, 20]$. $*$ denotes convolution with a blur kernel $K(N, \theta) = \frac{1}{N} R_\theta(\delta_c \otimes \mathbf{1}_N)$ constructed by rotating an impulse $\delta_c \otimes \mathbf{1}_N$ with random angle $R_\theta$ ($\theta \in [10, 80]$), using $\frac{1}{N}$ to normalize the kernel energy. Here, $N \in [3, 12]$ controls the blur level.

**Illumination-related Distortions.** Following the theoretical framework of Retinex, the formation of illumination-degraded images $D_{vi}^i$ is mathematically modeled as:

$$D_{vi}^i = \mathcal{P}_i(I_{vi}) = \frac{I_{vi}}{L} \cdot L^\gamma, \tag{4}$$

where $L$ is the illumination map estimated by LIME [5]. $\gamma \in [0.5, 3]$ controls the illumination level.

**Weather-related Distortions.** According to the methodologies in [21, 12], we employ the following formula to simulate weather-related degradations (*i.e.* rain and haze):

$$D_{vi}^w = \mathcal{P}_w(I_{vi}) = I_{vi} \cdot t + A(1 - t) + R, \tag{5}$$

where $t$ and $A$ denote the transmission map and atmospheric light, respectively. And $t$ is defined by the exponential decay of light, expressed as $t = e^{-\beta d(x)}$, where the haze density coefficient $\beta \in [0.5, 2.0]$. $d(x)$ refers to the scene depth, estimated by DepthAnything-V2 [40]. Besides, $A$ is constrained within $[0.3, 0.9]$ for realistic simulation.

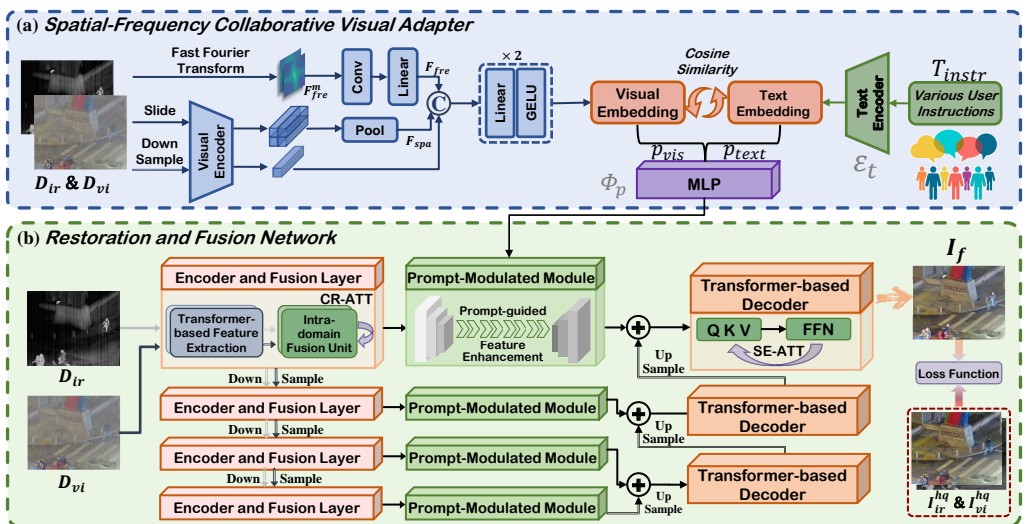

Figure 2: The overall framework of our controllable image fusion network.

Based on Eq. (1), we construct a multi-modal composite **D**egradation **D**ataset with four **L**evels (DDL-12), comprising 12 distinct degradations. We selecte $2,050$ high-quality clear images from RoadScene [36], LLVIP [7], and MSRS [29] datasets. Subsequently, the degraded imaging model is employed to synthesize degraded images with 12 types and 4 levels of degradation. Finally, the DDL-12 dataset contains approximately $48,000$ training image pairs and $4,800$ test image pairs.

## 4    Methodology

### 4.1    Problem Formulation

Given two source images $I_{ir} \in \mathbb{R}^{H \times W \times 1}$ and $I_{vi} \in \mathbb{R}^{H \times W \times 3}$, the typical fusion paradigm employs a network $\mathcal{N}_f$ to synthesize the fused image $I_f$, expressed as: $I_f = \mathcal{N}_f(I_{ir}, I_{vi})$. However, in complex scenarios, source images usually suffer from degradation interference, thus the advanced fusion paradigm must take both image restoration and fusion into account. While the concatenation approach is straightforward, it does not model recovery and fusion as an end-to-end optimization process, yielding suboptimal outcomes. As illustrated in Fig. 2, we propose a controllable paradigm that couples restoration and fusion with degradation prompts, termed ControlFusion, to overcome this limitation. On the one hand, the coupled approach enhances task synergy. On the other hand, leveraging degradation prompts to modulate the restoration and fusion process enables the network to adapt to diverse degradation distributions while meeting user-specific customization requirements. The proposed restoration and fusion paradigm can be defined as: $I_f = \mathcal{N}_{rf}(I_{ir}, I_{vi}, p \mid \Omega)$, where $\mathcal{N}_{rf}$ is a restoration and fusion network, $P$ denotes degradation prompts, and $\Omega$ indicates the degradation set. Our ControlFusion employs a two-stage optimization protocol. Stage I aligns textual prompts with visual embedding, while Stage II optimizes the overall framework.

### 4.2    Stage I: Textual-Visual Prompts Alignment

**Spatial-Frequency Collaborative Visual Adapter.** To achieve unified multimodal degradation representation, we first employ text semantics for explicit degradation modeling. Using a pre-trained CLIP [25] text encoder $\mathcal{E}_t$, the user instruction $T_{instr}$ is converted into text embedding: $p_{text} = \mathcal{E}_t(T_{instr})$. However, text-dependent models limit deployment flexibility. We therefore develop a spatial-frequency collaborative visual adapter (SFVA) to extract visual embeddings directly from images while maintaining semantic alignment with $p_{text}$.

As shown in Fig. 3, degraded images exhibit distinct spectral characteristics, indicating frequency features contain rich degradation priors. Our SFVA contains dual branches: In the frequency branch,

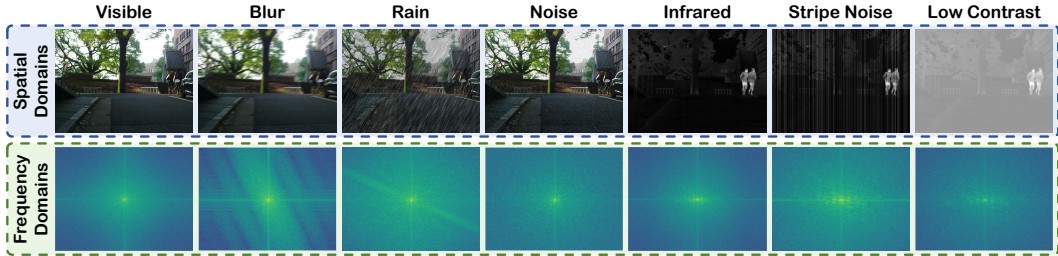

Figure 3: The visualization of various types of degradation in the spatial and frequency domains.

Fast Fourier Transform (FFT) and CNN extract frequency features:

$$F_{fre}^m = \sum_{x=0}^{W-1} \sum_{y=0}^{H-1} D_m(x,y)e^{-j2\pi\left(\frac{ux}{W}+\frac{vy}{H}\right)}, F_{fre} = \text{Linear}\left(\text{Conv}\left(\left[F_{fre}^{ir}, F_{fre}^{vi}\right]\right)\right), \quad (6)$$

where $[\cdot, \cdot]$ denotes channel-wise concatenation. The spatial branch employs cropping/downsampling augmentation and CNN to extract spatial features $F_{spa}$.

The fused visual embedding $p_{vis}$ is obtained through concatenation and linear projection of $F_{fre}$ and $F_{spa}$. To ensure semantic consistency between visual/text embeddings, we apply MSE loss $\mathcal{L}_{mse}$ and cosine similarity loss $\mathcal{L}_{cos}$:

$$\mathcal{L}_I = \lambda_1 \underbrace{\|p_{vis} - p_{text}\|^2}_{\mathcal{L}_{\text{mse}}} + \lambda_2 \underbrace{\left(1 - \frac{p_{vis} \cdot p_{text}}{\|p_{vis}\|\|p_{text}\|}\right)}_{\mathcal{L}_{\text{cos}}}, \quad (7)$$

where $\lambda_1$ and $\lambda_2$ balance loss components. This design enables automatic degradation-aware adaptation while preserving text-aligned semantics.

### 4.3 Stage II: Prompt-modulated Restoration and Fusion

#### 4.3.1 Network Architectures

**Feature Encoding and Fusion Layer.** As shown Fig. 2, we devise the hierarchical transformer-based encoders to extract multi-scale feature representations from degraded infrared ($D_{ir}$) and visible ($D_{vi}$) images separately, which is formulated as: $\{F_{ir}, F_{vi}\} = \mathcal{E}_{ir}(D_{ir}), \mathcal{E}_{vi}(D_{vi})$, where $\mathcal{E}_{ir}$ and $\mathcal{E}_{vi}$ are infrared and visible image encoders.

Furthermore, to achieve comprehensive cross-modal feature integration, we design the intra-domain fusion unit, where the cross-attention (CR-ATT) mechanism is employed to facilitate interaction between features across modalities. Specifically, the linear transformation functions $\mathcal{F}_{ir}^{qkv}$ and $\mathcal{F}_{vi}^{qkv}$ project $F_{ir}$ and $F_{vi}$ into their corresponding $Q$, $K$, and $V$, expressed as:

$$\{Q_{ir}, K_{ir}, V_{ir}\} = \mathcal{F}_{ir}^{qkv}(F_{ir}), \{Q_{vi}, K_{vi}, V_{vi}\} = \mathcal{F}_{vi}^{qkv}(F_{vi}). \quad (8)$$

Subsequently, we swap the queries $Q$ of complementary modalities to promote spatial interaction:

$$F_f^{ir} = \text{softmax}\left(\frac{Q_{vi}K_{ir}}{\sqrt{d_k}}\right)V_{ir}, F_f^{vi} = \text{softmax}\left(\frac{Q_{ir}K_{vi}}{\sqrt{d_k}}\right)V_{vi}, \quad (9)$$

where $d_k$ is scaling factor. We concatenate $F_f^{ir}$ and $F_f^{vi}$ to get fusion features: $F_f = [F_f^{ir}, F_f^{vi}]$.

**Prompt-modulated Module and Image Decoder**. To dynamically adapt fusion feature distributions to degradation patterns and degrees, we propose a Prompt-Modulated Module (PMM) for robust feature enhancement. First, sequential MLPs ($\Phi_p$) derive distributional optimization parameters: $[\gamma_p, \beta_p] = \Phi_p(p)$, where $p \in \{p_{\text{vis}}, p_{\text{text}}\}$. Then, $\gamma_p$ and $\beta_p$ are applied for feature scaling and bias shifting in a residual manner: $\hat{F}_f = (1 + \gamma_p) \odot F_f + \beta_p$, where $\odot$ denotes the Hadamard product, and $\hat{F}_f$ indicates enhanced features incorporating degradation prompts. We further deploy a series of Transformer-based decoder $\mathcal{D}_f$ with the self-attention to progressively reconstruct the fused image $I_f = \mathcal{D}_f(\hat{F}_f)$. In particular, PMM and $\mathcal{D}_f$ are tightly coupled in a multi-stage process, effectively incorporating fusion features and degradation prompts to synthesize the desired image.

### 4.3.2 Loss Functions

Following the typical fusion paradigm [20], we introduce the intensity loss, structural similarity (SSIM) loss, maximum gradient loss, and color consistency loss to constrain the training of Stage II.

The intensity loss $\mathcal{L}_{int}$ maximizes the target prominence of fusion results, defined as:

$$\mathcal{L}_{int} = \frac{1}{HW}\|I_f - \max(I_{ir}^{hq}, I_{vi}^{hq})\|_1, \tag{10}$$

where $I_{ir}^{hq}$ and $I_{vi}^{hq}$ are the high-quality source images.

The structural similarity loss $\mathcal{L}_{ssim}$ ensures the fused image maintains structural consistency with the high-quality source images, preserving essential structural information, and is formulated as:

$$\mathcal{L}_{ssim} = 2 - (\text{SSIM}(I_f, I_{ir}^{hq}) + \text{SSIM}(I_f, I_{vi}^{hq})). \tag{11}$$

The maximum gradient loss $\mathcal{L}_{grad}$ maximizes the retention of key edge information from both source images, generating fusion results with clearer textures, formulated as:

$$\mathcal{L}_{grad} = \frac{1}{HW}\|\nabla I_f - \max(\nabla I_{ir}^{hq}, \nabla I_{vi}^{hq})\|_1, \tag{12}$$

where $\nabla$ denotes the Sobel operator. Moreover, the color consistency loss $\mathcal{L}_{color}$ ensures that the fusion results maintain color consistency with the visible image. We convert the image to YCbCr space and minimize the distance between the Cb and Cr channels, expressed as:

$$\mathcal{L}_{color} = \frac{1}{HW}\|\mathcal{F}_{CbCr}(I_f) - \mathcal{F}_{CbCr}(I_{vi}^{hq})\|_1, \tag{13}$$

where $\mathcal{F}_{CbCr}$ denotes the transfer function of RGB to CbCr.

Finally, the total loss $\mathcal{L}_{II}$ for Stage II is the weighted sum of the aforementioned losses:

$$\mathcal{L}_{II} = \alpha_{int} \cdot \mathcal{L}_{int} + \alpha_{ssim} \cdot \mathcal{L}_{ssim} + \alpha_{grad} \cdot \mathcal{L}_{grad} + \alpha_{color} \cdot \mathcal{L}_{color}, \tag{14}$$

where $\alpha_{int}$, $\alpha_{ssim}$, $\alpha_{grad}$, and $\alpha_{color}$ are hyper-parameters.

## 5 Experiments

### 5.1 Implementation and Experimental Configurations

Our image restoration and fusion network is built on a four-stage encoder–decoder architecture, with channel dimensions increasing from 48 to 384, specifically configured as $[48, 96, 192, 384]$. The model is trained on the proposed DDL-12 dataset. During training, $224 \times 224$ patches are randomly cropped as inputs, with a batch size of 12 over 100 epochs. Optimization is performed using AdamW, starting with a learning rate of $1 \times 10^{-3}$ and decayed to $1 \times 10^{-5}$ via a cosine annealing schedule. For the loss configuration, $\lambda_1$ and $\lambda_2$ are set with a weight ratio of $1 : 3$, and $\alpha_{int}$, $\alpha_{ssim}$, $\alpha_{grad}$, and $\alpha_{color}$ are assigned values of $8 : 1 : 10 : 12$, respectively.

We introduce text prompts to enable the unified network to effectively handle diverse and complex degradations. Each prompt specifies the affected *modality*, the *degradation type*, and its *severity*, enabling user-controllable flexibility. For example, a typical prompt for a single degradation is: *We are performing infrared and visible image fusion, where the **modality** suffers from a grade-**severity degradation type***. For composite degradations, we extend this template to specify multiple modality–degradation pairs, e.g., *We are performing infrared and visible image fusion. Please handle a grade-**severity-A degradation type-A** in the **modality-A**, and a grade-**severity-B degradation type-B** in the **modality-B***. Details of the prompt construction paradigm are provided in the Appendix.

We compare our method with seven SOTA fusion methods, *i.e.*, DDFM [50], DRMF [31], EMMA [51], LRRNet [11], SegMiF [16], Text-IF [39], and Text-DiFuse [42]. Firstly, we evaluate the fusion performance on four widely used datasets, *i.e.*, MSRS [29], LLVIP [7], RoadScene [35], and FMB [16]. The test set sizes for MSRS, LLVIP, RoadScene, and FMB are 361, 50, 50, and 50, respectively. Four metrics, *i.e.*, EN, SD, VIF, and $Q_{abf}$, are used to quantify fusion performance. Additionally, we evaluate the restoration and fusion performance across various degradations, including low-contrast, random noise, and stripe noise in infrared images (IR), as well as blur, rain, over-exposure, and low light in visible images (VI). Additionally, we evaluate its effectiveness in handling composite degradation scenarios. Each degradation scenario includes 100 image pairs from DDL-12 dataset. We use CLIP-IQA [34], MUSIQ [9], TReS [3], EN, and SD to quantify both restoration and fusion performance. All experiments are conducted on NVIDIA RTX 4090 GPUs with an Intel(R) Xeon(R) Platinum 8180 CPU (2.50 GHz) using the PyTorch framework.

Table 1: Quantitative comparison results on typical fusion datasets. The best and second-best results are highlighted in Red and Purple.

| Methods | MSRS | | | | LLVIP | | | | RoadScene | | | | FMB | | | |
|---|---|---|---|---|---|---|---|---|---|---|---|---|---|---|---|---|
| | EN | SD | VIF | Qabf | EN | SD | VIF | Qabf | EN | SD | VIF | Qabf | EN | SD | VIF | Qabf |
| DDFM | 6.431 | 47.815 | 0.844 | 0.643 | 6.914 | 48.556 | 0.693 | 0.517 | 6.994 | 47.094 | 0.775 | 0.595 | 6.426 | 40.597 | 0.495 | 0.442 |
| DRMF | 6.268 | 45.117 | 0.669 | 0.550 | 6.901 | 50.736 | 0.786 | 0.626 | 6.231 | 44.221 | 0.728 | 0.527 | 6.842 | 41.816 | 0.578 | 0.372 |
| EMMA | 6.747 | 52.753 | 0.886 | 0.605 | 6.366 | 47.065 | 0.743 | 0.547 | 6.959 | 46.749 | 0.698 | 0.664 | 6.788 | 38.174 | 0.542 | 0.436 |
| LRRNet | 6.761 | 49.574 | 0.713 | 0.667 | 6.191 | 48.336 | 0.864 | 0.575 | 7.185 | 46.400 | 0.756 | 0.658 | 6.432 | 48.154 | 0.501 | 0.368 |
| SegMiF | 7.006 | 57.073 | 0.764 | 0.586 | 7.260 | 45.892 | 0.539 | 0.459 | 6.736 | 48.975 | 0.629 | 0.584 | 6.363 | 47.398 | 0.539 | 0.482 |
| Text-IF | 6.619 | 55.881 | 0.753 | 0.656 | 6.364 | 49.868 | 0.859 | 0.566 | 6.836 | 47.596 | 0.634 | 0.609 | 7.397 | 47.726 | 0.568 | 0.528 |
| Text-DiFuse | 6.990 | 56.698 | 0.850 | 0.603 | 7.546 | 55.725 | 0.883 | 0.659 | 6.826 | 50.230 | 0.683 | 0.662 | 6.888 | 49.558 | 0.793 | 0.653 |
| ControlFusion | 7.340 | 60.360 | 0.927 | 0.718 | 7.354 | 56.631 | 0.968 | 0.738 | 7.421 | 51.759 | 0.817 | 0.711 | 7.036 | 50.905 | 0.872 | 0.730 |

Table 2: Quantitative comparison results under different degradation scenarios with pre-enhancement.

| Methods | VI (Blur) | | | | VI (Rain) | | | | VI (Low light, LL) | | | | VI (Over-exposure, OE) | | | |
|---|---|---|---|---|---|---|---|---|---|---|---|---|---|---|---|---|
| | CLIP-IQA | MUSIQ | TReS | SD | CLIP-IQA | MUSIQ | TReS | SD | CLIP-IQA | MUSIQ | TReS | SD | CLIP-IQA | MUSIQ | TReS | SD |
| DDFM | 0.141 | 39.421 | 39.047 | 35.411 | 0.191 | 38.836 | 46.285 | 36.376 | 0.156 | 39.495 | 41.782 | 31.759 | 0.143 | 43.167 | 43.440 | 32.099 |
| DRMF | 0.128 | 40.739 | 40.968 | 40.722 | 0.174 | 48.164 | 48.565 | 41.174 | 0.143 | 41.428 | 37.947 | 38.287 | 0.190 | 48.334 | 42.582 | 44.256 |
| EMMA | 0.131 | 43.472 | 41.744 | 42.553 | 0.138 | 45.824 | 44.916 | 43.378 | 0.158 | 39.674 | 44.827 | 40.857 | 0.180 | 46.731 | 47.616 | 40.242 |
| LRRNet | 0.163 | 42.981 | 37.268 | 45.389 | 0.185 | 43.291 | 41.891 | 46.285 | 0.164 | 40.486 | 34.836 | 41.639 | 0.160 | 42.548 | 48.414 | 42.190 |
| SegMiF | 0.152 | 43.005 | 43.516 | 44.000 | 0.195 | 40.528 | 49.094 | 44.274 | 0.177 | 44.073 | 48.376 | 44.829 | 0.166 | 49.132 | 38.019 | 38.484 |
| Text-IF | 0.164 | 44.801 | 46.542 | 48.401 | 0.164 | 41.287 | 47.380 | 49.298 | 0.163 | 41.096 | 49.174 | 47.257 | 0.172 | 40.298 | 45.599 | 47.330 |
| Text-DiFuse | 0.172 | 44.958 | 47.699 | 46.376 | 0.173 | 39.243 | 50.017 | 47.297 | 0.192 | 46.734 | 50.126 | 49.883 | 0.183 | 39.095 | 49.596 | 50.279 |
| ControlFusion | 0.184 | 47.849 | 50.240 | 50.287 | 0.196 | 52.319 | 52.465 | 50.901 | 0.183 | 48.420 | 51.072 | 53.787 | 0.191 | 50.301 | 52.961 | 54.218 |

| Methods | VI (Rain and Haze, RH) | | | | IR (Low-contrast, LC) | | | | IR (Random noise, RN) | | | | IR (Stripe noise, SN) | | | |
|---|---|---|---|---|---|---|---|---|---|---|---|---|---|---|---|---|
| | CLIP-IQA | MUSIQ | TReS | EN | CLIP-IQA | MUSIQ | TReS | SD | CLIP-IQA | MUSIQ | TReS | EN | CLIP-IQA | MUSIQ | TReS | EN |
| DDFM | 0.158 | 43.119 | 44.095 | 6.897 | 0.172 | 44.490 | 44.545 | 37.452 | 0.186 | 34.059 | 32.807 | 6.029 | 0.209 | 48.479 | 32.377 | 6.399 |
| DRMF | 0.171 | 45.524 | 43.693 | 6.230 | 0.208 | 43.982 | 45.561 | 40.315 | 0.192 | 44.617 | 44.085 | 5.744 | 0.187 | 44.714 | 43.650 | 6.354 |
| EMMA | 0.169 | 39.092 | 46.046 | 6.517 | 0.154 | 48.724 | 43.113 | 53.861 | 0.172 | 39.666 | 44.466 | 6.184 | 0.153 | 42.802 | 44.239 | 6.382 |
| LRRNet | 0.170 | 48.571 | 48.973 | 7.363 | 0.151 | 48.718 | 45.266 | 51.605 | 0.158 | 48.274 | 37.090 | 7.295 | 0.137 | 46.511 | 36.610 | 7.702 |
| SegMiF | 0.147 | 46.139 | 45.019 | 7.281 | 0.160 | 44.659 | 51.427 | 44.448 | 0.180 | 42.664 | 39.496 | 6.669 | 0.169 | 49.887 | 39.247 | 6.855 |
| Text-IF | 0.178 | 50.568 | 50.271 | 6.956 | 0.187 | 49.299 | 49.266 | 47.032 | 0.169 | 46.647 | 48.491 | 6.256 | 0.161 | 49.019 | 47.755 | 6.085 |
| Text-DiFuse | 0.175 | 52.788 | 53.073 | 7.470 | 0.165 | 50.092 | 51.715 | 39.429 | 0.203 | 49.278 | 49.479 | 6.982 | 0.194 | 51.762 | 49.288 | 7.089 |
| ControlFusion | 0.189 | 54.287 | 54.465 | 7.891 | 0.196 | 51.986 | 52.846 | 57.827 | 0.189 | 50.711 | 51.668 | 7.724 | 0.200 | 50.097 | 50.264 | 7.619 |

| Methods | VI (OE) and IR (LC) | | | | VI(Low light and Noise, LN) | | | | VI (RH) and IR (RN) | | | | VI (LL) and IR (SN) | | | |
|---|---|---|---|---|---|---|---|---|---|---|---|---|---|---|---|---|
| | CLIP-IQA | MUSIQ | TReS | SD | CLIP-IQA | MUSIQ | TReS | EN | CLIP-IQA | MUSIQ | TReS | SD | CLIP-IQA | MUSIQ | TReS | EN |
| DDFM | 0.168 | 43.814 | 41.894 | 36.095 | 0.172 | 48.293 | 31.791 | 6.298 | 0.151 | 33.440 | 32.134 | 37.342 | 0.189 | 36.433 | 42.630 | 5.776 |
| DRMF | 0.184 | 42.399 | 39.374 | 40.847 | 0.201 | 44.363 | 43.063 | 5.875 | 0.174 | 43.663 | 43.858 | 37.997 | 0.142 | 38.241 | 41.049 | 5.280 |
| EMMA | 0.130 | 39.892 | 42.076 | 43.362 | 0.174 | 42.201 | 43.382 | 5.838 | 0.165 | 39.146 | 44.458 | 51.205 | 0.130 | 37.367 | 43.888 | 6.318 |
| LRRNet | 0.136 | 47.209 | 42.636 | 46.684 | 0.144 | 46.386 | 35.779 | 7.306 | 0.128 | 47.954 | 36.831 | 49.917 | 0.154 | 38.426 | 35.970 | 7.007 |
| SegMiF | 0.114 | 44.021 | 42.256 | 33.647 | 0.136 | 49.178 | 38.570 | 5.819 | 0.147 | 42.354 | 39.156 | 31.717 | 0.151 | 41.287 | 37.079 | 6.767 |
| Text-IF | 0.174 | 48.808 | 47.998 | 48.848 | 0.217 | 48.100 | 47.510 | 5.204 | 0.158 | 45.821 | 47.626 | 46.543 | 0.140 | 41.429 | 46.220 | 5.525 |
| Text-DiFuse | 0.131 | 49.021 | 50.980 | 47.640 | 0.185 | 50.775 | 48.610 | 6.440 | 0.181 | 48.645 | 48.937 | 38.808 | 0.161 | 47.734 | 48.448 | 6.738 |
| ControlFusion | 0.187 | 50.479 | 50.298 | 50.955 | 0.225 | 49.333 | 49.513 | 7.111 | 0.179 | 50.107 | 51.091 | 55.417 | 0.167 | 50.632 | 48.971 | 7.055 |

## 5.2 Fusion Performance Comparison

**Comparison without Pre-enhancement.** Table 1 summarizes the quantitative assessment across benchmark datasets. Notably, our method attains SOTA performance in SD, VIF, and $Q_{abf}$ metrics, showcasing its exceptional ability to maintain structural integrity and edge details. The EN metric remains competitive, indicating that our fusion results encapsulate abundant information.

**Comparison with Pre-enhancement.** For a fair comparison, we exclusively use the visual prompts to characterize degradation in the quantitative comparison. Meanwhile, we employ several SOTA image restoration algorithms as preprocessing steps to address specific degradations. Specifically, OneRestore [6] is used for weather-related degradations, Instruct-IR [2] for illumination-related degradations, and DA-CLIP [18] and WDNN [4] for sensor-related degradations. For composite degradations, we select the best-performing method from these algorithms.

The quantitative results in Tab. 2 show that ControlFusion achieves superior CLIP-IQA, MUSIQ, and TReS scores across most degradation scenarios, demonstrating a strong capability in degradation mitigation and complementary context aggregation. Additionally, no-reference fusion metrics (SD, EN) show that ControlFusion performs on par with SOTA image fusion methods. Qualitative comparisons in Fig. 4 indicate that ControlFusion not only excels in addressing single-modal degradation but also effectively tackles more challenging single- and multi-modal composite degradations. In particular, when rain and haze coexist in the visible image, ControlFusion successfully perceives these degradations and eliminates their effects. In multi-modal composite degradation scenarios, it leverages degradation prompts to adjust the distribution of fusion features, achieving high-quality restoration and fusion. Extensive experiments demonstrate that ControlFusion exhibits significant advantages in complex degradation scenarios.

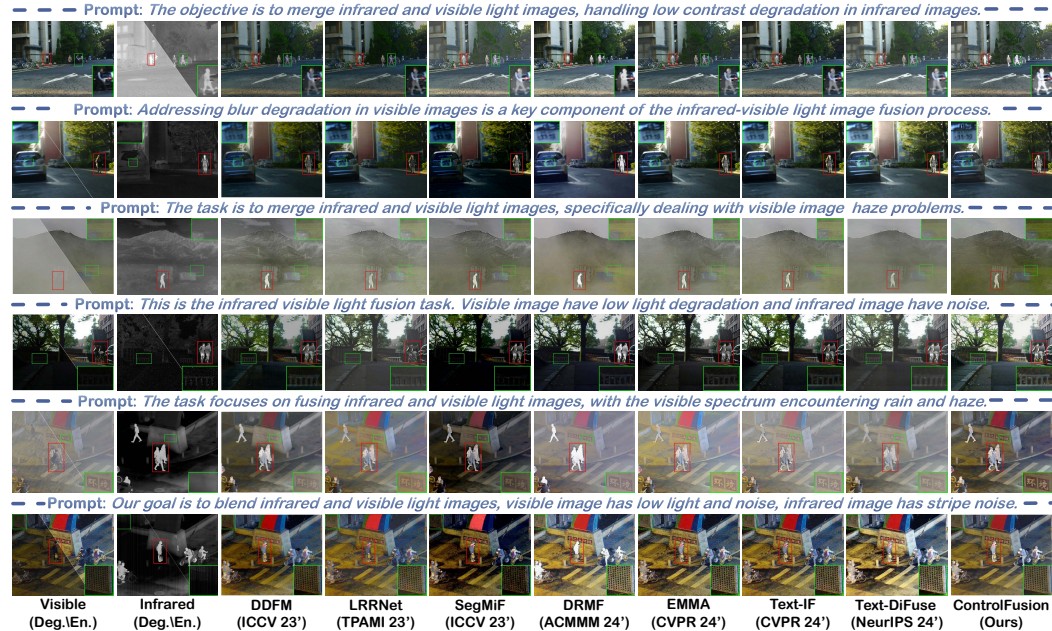

**Prompt:** *The objective is to merge infrared and visible light images, handling low contrast degradation in infrared images.*

**Prompt:** *Addressing blur degradation in visible images is a key component of the infrared-visible light image fusion process.*

**Prompt:** *The task is to merge infrared and visible light images, specifically dealing with visible image haze problems.*

**Prompt:** *This is the infrared visible light fusion task. Visible image have low light degradation and infrared image have noise.*

**Prompt:** *The task focuses on fusing infrared and visible light images, with the visible spectrum encountering rain and haze.*

**Prompt:** *Our goal is to blend infrared and visible light images, visible image has low light and noise, infrared image has stripe noise.*

| Visible (Deg.\En.) | Infrared (Deg.\En.) | DDFM (ICCV 23') | LRRNet (TPAMI 23') | SegMiF (ICCV 23') | DRMF (ACMMM 24') | EMMA (CVPR 24') | Text-IF (CVPR 24') | Text-DiFuse (NeurIPS 24') | ControlFusion (Ours) |

Figure 4: Visualization of fusion results under different degradation scenarios.

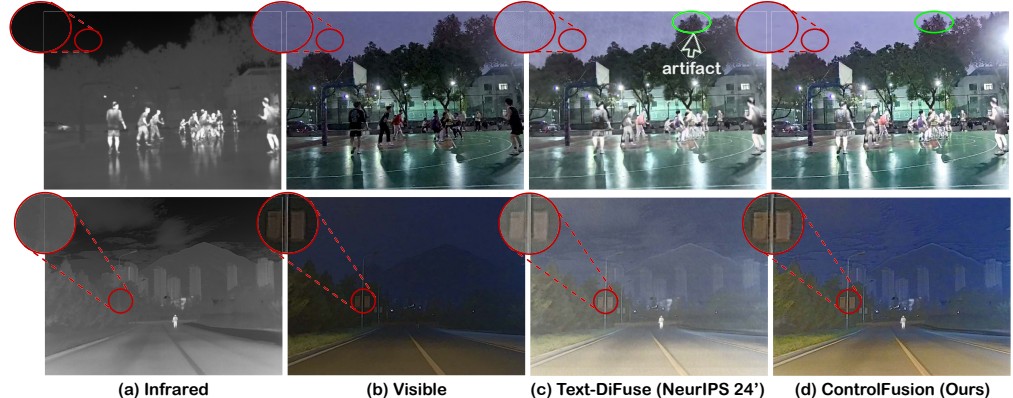

| (a) Infrared | (b) Visible | (c) Text-DiFuse (NeurIPS 24') | (d) ControlFusion (Ours) |

Figure 5: Generalization results under real-world degradation scenarios.

## 5.3 Extended Experiments

**Real-world Generalization.** As shown in Tab. 3, we report comparative results on nighttime scenes from MSRS and the real weather-degraded benchmark AWMM-100k [13]. Our method achieves the best performance on CLIP-IQA, TRes, and SD, while slightly trailing Text-DiFuse on MUSIQ . On AWMM-100k, our approach consistently outperforms competing methods across all evaluation metrics under real-world weather degradation conditions.

Table 3: Quantitative comparison on MSRS nighttime scenes and AWMM-100k with real-world weather degradations.

| Methods | MSRS (Nighttime) | | | | AWMM-100k (Weather Degradations) | | | |
|---|---|---|---|---|---|---|---|---|
| | CLIP-IQA | MUSIQ | TReS | SD | CLIP-IQA | MUSIQ | TReS | SD |
| DDFM | 0.104 | 25.034 | 19.404 | 33.935 | 0.227 | 41.919 | 53.858 | 36.010 |
| DRMF | 0.102 | 27.518 | 16.052 | 46.718 | 0.232 | 50.054 | 56.671 | 39.735 |
| EMMA | 0.087 | 26.713 | 18.511 | 42.659 | 0.253 | 47.161 | 49.722 | 40.398 |
| LRRNet | 0.098 | 27.042 | 16.909 | 42.692 | 0.192 | 46.751 | 50.707 | 37.190 |
| SegMiF | 0.093 | 27.405 | 21.079 | 44.315 | 0.257 | 50.712 | 54.310 | 33.311 |
| Text-IF | 0.101 | 28.155 | 20.640 | 49.752 | 0.213 | 45.633 | 51.747 | 32.086 |
| Text-DiFuse | 0.116 | **28.620** | 18.967 | 43.728 | 0.233 | 46.901 | 57.708 | 41.530 |
| ControlFusion | **0.141** | 28.457 | **21.607** | **52.933** | **0.280** | **57.513** | **60.302** | **44.244** |

In addition to quantitative comparisons, Fig. 5 presents visual examples demonstrating our method's strong generalization ability in removing composite degradations in real scenes. The top image shows typical data collected by our multimodal sensors under challenging low-light and noisy conditions,

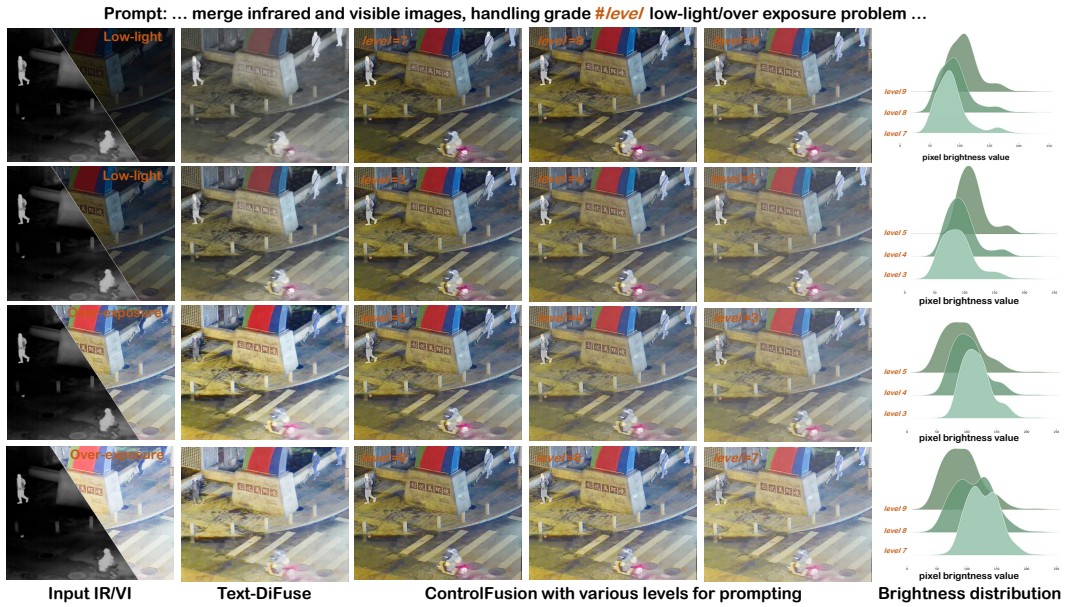

Prompt: … merge infrared and visible images, handling grade *#level* low-light/over exposure problem …

**Input IR/VI** **Text-DiFuse** **ControlFusion with various levels for prompting** **Brightness distribution**

Figure 6: Fusion results with various level prompts.

Table 4: Computational efficiency of image restoration and fusion algorithms on $640 \times 480$ resolution.

| Image Restoration | | | | Image Fusion | | | | | | | |
|---|---|---|---|---|---|---|---|---|---|---|---|
| Methods | Parm.(M) | Flops(G) | Time(s) | Methods | Parm.(M) | Flops(G) | Time(s) | Methods | Parm.(M) | Flops(G) | Time(s) |
| InstructIR | 15.94 | 151.54 | 0.038 | DDFM* | 552.70 | 5220.50 | 34.511 | SegMiF* | 45.64 | 707.39 | 0.371 |
| DA-CLIP | 295.19 | 1214.55 | 13.578 | DRMF | 5.05 | 121.10 | 0.842 | Text-IF | 152.44 | 1518.90 | 0.157 |
| WDNN | 0.01 | 1.11 | 0.004 | EMMA* | 1.52 | 41.54 | 0.048 | Text-DiFuse | 157.35 | 4872.00 | 12.903 |
| OneRestore | 18.12 | 114.56 | 0.024 | LRRNet* | 0.05 | 14.17 | 0.096 | ControlFusion | 182.54 | 1622.00 | 0.183 |

while the bottom image illustrates the robustness of our method in handling extremely low-light scenarios. Additional qualitative results are provided in the Appendix.

**Flexible Degree Control.** We employ numerical values to quantify the severity of degradation in design, where higher numbers indicate more severe degradation, enabling precise continuous control. It should be noted that while only four anchor points (1, 4, 7, and 10) were used during training, the model demonstrates remarkable generalization capability to intermediate degrees during testing, owing to the inherent encoding characteristics of CLIP. As shown in Fig. 6, with degradation level prompts, our ControlFusion can adapt to varying degrees of degradation and deliver satisfactory fusion results. Moreover, fusion results generated using adjacent degradation-level prompts exhibit subtle yet perceptible differences, allowing users to obtain customized outcomes through their specific prompts. More visualization results are presented in the Appendix.

**Computational Efficiency.** Table 4 details the computational costs for various methods. While standalone fusion models such as EMMA, LRRNet, and SegMiF, which are marked with *, appear more lightweight, they necessitate a separate, computationally intensive restoration stage to form a practical pipeline. When the overhead of this two-stage process is factored in, the efficiency advantages of our unified model becomes evident. Moreover, our model's efficiency remains highly competitive with other joint restoration-fusion approaches like DRMF, Text-DiFuse, and Text-IF.

**Object Detection.** We evaluate object detection performance on LLVIP to assess the fusion quality, using the re-trained YOLOv8 [26]. Quantitative results are presented in Tab. 5. Our results enable the detector to identify all pedestrians with higher confidence, achieving the highest mAP@0.5-0.95. Visualizations are in the Appendix.

Table 5: Quantitative comparison of object detection.

| Methods | Prec. | Recall | AP@0.50 | AP@0.75 | mAP@0.5:0.95 |
|---|---|---|---|---|---|
| DDFM | 0.947 | 0.848 | 0.911 | 0.655 | 0.592 |
| DRMF | 0.958 | 0.851 | 0.937 | 0.672 | 0.607 |
| EMMA | 0.942 | 0.872 | 0.927 | 0.647 | 0.598 |
| LRRNet | 0.939 | 0.878 | 0.933 | 0.672 | 0.608 |
| SegMiF | 0.965 | **0.896** | 0.931 | **0.690** | 0.603 |
| Text-IF | 0.959 | 0.892 | 0.939 | 0.655 | 0.601 |
| Text-DiFuse | 0.961 | 0.885 | 0.941 | 0.656 | 0.606 |
| ControlFusion | **0.971** | 0.889 | **0.949** | 0.685 | **0.609** |

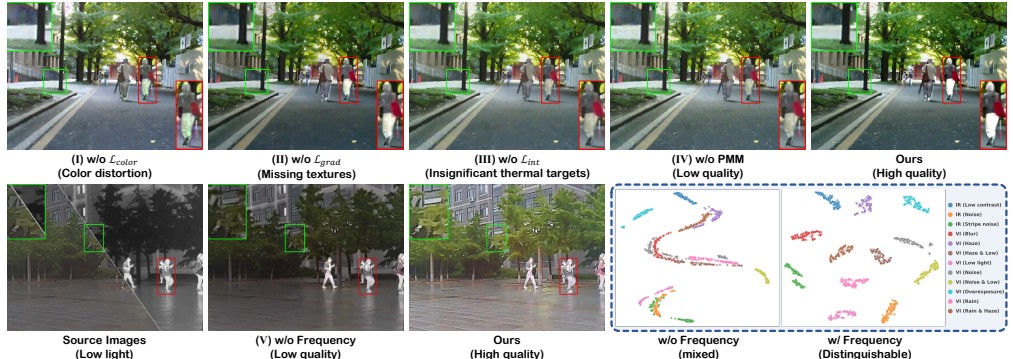

Figure 7: Visual results of ablation studies under degradation scenarios.

Table 6: Quantitative results of the ablation studies.

| Configs | VI(LL & Noise) | | | | VI(OE) and IR(LC) | | | | VI (RH) and IR(Noise) | | | |
|---|---|---|---|---|---|---|---|---|---|---|---|---|
| | CLIP-IQA | MUSIQ | TReS | EN | CLIP-IQA | MUSIQ | TReS | SD | CLIP-IQA | MUSIQ | TReS | SD |
| I | 0.132 | 42.424 | 42.839 | 4.788 | 0.166 | 41.983 | 42.841 | 39.917 | 0.147 | 43.619 | 47.932 | 48.935 |
| II | 0.152 | 45.582 | 45.358 | 5.855 | 0.151 | 43.646 | 42.002 | 39.208 | 0.167 | 47.862 | 45.007 | 44.347 |
| III | 0.154 | 46.571 | 44.495 | 5.013 | 0.155 | 44.286 | 44.561 | 42.068 | 0.156 | 43.007 | 43.816 | 41.544 |
| IV | 0.129 | 38.960 | 41.310 | 5.414 | 0.172 | 41.743 | 39.125 | 38.748 | 0.118 | 48.882 | 46.245 | 45.910 |
| V | 0.173 | 45.281 | 46.291 | 6.279 | 0.181 | 45.386 | 47.519 | 46.860 | 0.149 | 46.714 | 48.094 | 46.950 |
| **Ours** | **0.225** | **49.333** | **49.513** | **7.111** | **0.187** | **50.479** | **50.298** | **50.955** | **0.179** | **50.107** | **51.091** | **55.417** |

**Ablation Studies.** To verify the effectiveness of components, we conduct detailed ablation studies, involves: (I) **w/o** $\mathcal{L}_{color}$, (II) **w/o** $\mathcal{L}_{grad}$, (III) **w/o** $\mathcal{L}_{int}$, (IV) **w/o PMM**, and (V) **w/o frequency** branch. As shown in Fig. 7, removing any loss or module significantly impacts the fusion quality. Specifically, without $\mathcal{L}_{color}$, color distortion occurs, while removing $\mathcal{L}_{grad}$ leads to the loss of essential texture information. Excluding $\mathcal{L}_{int}$ fails to highlight thermal targets, and removing PMM diminishes the ability to address composite degradation. Additionally, removing the frequency branch from SFVA causes visual prompt confusion. The t-SNE results further validate the importance of frequency priors. The quantitative results in Tab. 6 also confirm that each design element is crucial for enhancing fusion performance.

**Discussions and Limitations.** To mitigate the gap between simulated and real-world data, our method constructs training samples using the degradation imaging model. An alternative strategy is test-time adaptation, in which part of the model is fine-tuned on the test set to better accommodate new data distributions. It should be noted that the degradation imaging model is specifically designed for infrared and visible image fusion and is difficult to generalize to other fusion tasks, such as medical image fusion and multi-focus image fusion.

# 6   Conclusion

This work proposes a controllable framework for image restoration and fusion leveraging language-visual prompts. Initially, we develop a physics-driven degraded imaging model to bridge the domain gap between synthetic data and real-world images, providing a solid foundation for addressing composite degradations. Moreover, we devise a prompt-modulated network that adaptively adjusts the fusion feature distribution, enabling robust feature enhancement based on degradation prompts. Prompts can either come from text instructions to support user-defined control or be extracted from source images using a spatial-frequency visual adapter embedded with frequency priors, facilitating automated deployment. Extensive experiments demonstrate that our method excels in handling real-world and composite degradations, showing strong robustness across various degradation levels.

# 7   Acknowledgement

This work was supported by NSFC (62276192).

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

# A  Appendix

## A.1  Prompt Design Details

We provide additional details of the prompt construction paradigm to complement the main text.

**Prompt components.** Each prompt specifies three key components: (i) the affected *modality* (infrared or visible), (ii) the *degradation type* (*e.g.*, rain, haze, low-light, overexposure, random noise, stripe noise, low-contrast, etc.), and (iii) the *severity* level (an integer from 1 to 10).

**Severity levels.** To characterize degradation severity, we define a scale from 1 to 10, with representative anchor points as follows:

- **Level 1**: barely perceptible degradation,
- **Level 4**: degradation begins to interfere with human scene understanding,
- **Level 7**: degradation severely hinders human perception of the scene,
- **Level 10**: most useful information is completely obscured.

Intermediate levels can be interpolated between these anchor points.

**Prompt templates.** A typical template for a single degradation is structured as: We are performing infrared and visible image fusion, where the *modality* suffers from a grade-*severity degradation type*.

For composite degradations, two extensions are designed: 1) We are performing infrared and visible image fusion. Please handle a grade-*severity-A degradation type-A* in the *modality-A*, and a grade-*severity-B degradation type-B* in the *modality-B*; 2) We are performing infrared and visible image fusion. Please address level-*severity degradation type-A* and *degradation type-B* in the *modality*.

These templates flexibly represent both intra- and inter-modal combinations. For instance, an instantiation could be: We are performing infrared and visible image fusion. Please handle a grade-6 low-light in the visible modality, and a grade-8 stripe noise in the infrared modality.

**Linguistic diversity.** To enhance robustness, we curated over 100 unique textual formulations for each task and its associated parameters. This prompt rephrasing strategy augments linguistic variability and improves the model's generalization to diverse prompt styles during inference.

Additionally, when users are uncertain about severity, the proposed SFVA module can automatically perceive both degradation type and severity from the input. This design balances user-controllable flexibility with automated practicality.

## A.2  Visual Embedding versus Textual Embedding

From Fig. 8, we can find that both visual and textual prompts assist $N_{rf}$ in restoring the scene illumination. Besides, the residual plots between the two are not obvious, which indicates that our SFVA successfully captures the type and extent of degradation.This capability is crucial for automatic deployment, as it allows the model to adaptively adjust its restoration strategy with fewer manual lintervention or hyperparameters tuning for different degradation scenarios.

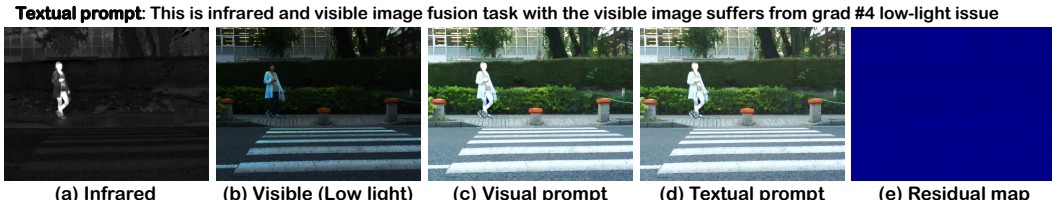

Figure 8: Comparison of visual and textual embedding.

## A.3  Flexible Degree Control

In order to verify the effectiveness of the proposed flexible degree control, we perform different degrees of prompts on a pair of low-light images. The results are shown in Fig. 9 (b). We splice all the results to obtain images with continuously changing brightness in different regions, and count the

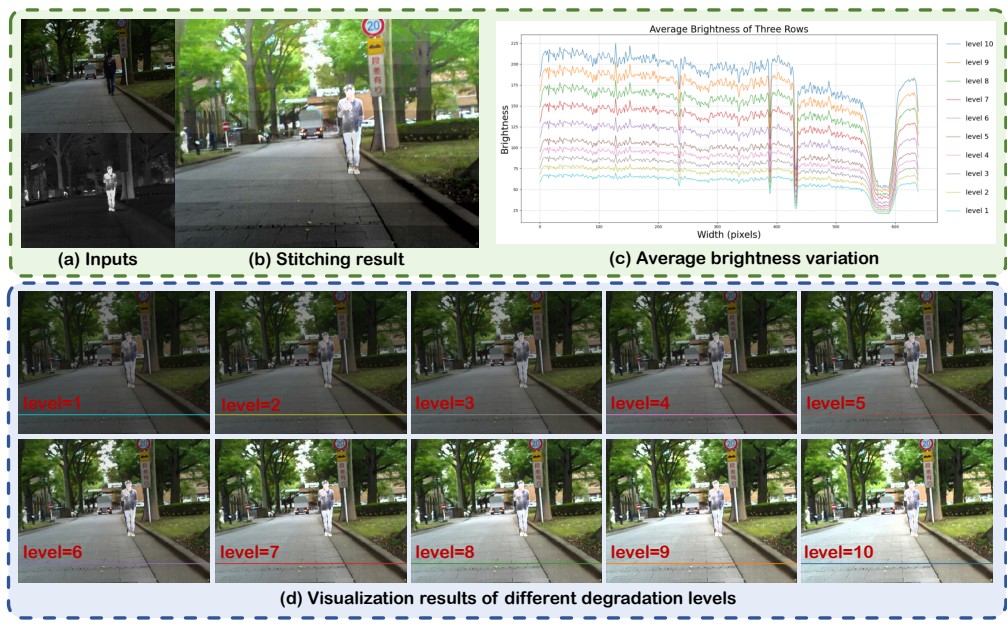

Figure 9: Fusion results with various level prompts.

brightness of the same region of these images. The results are shown in Fig. 9 (c). This shows that our degree control achieves fine-grained regulation.

## A.4 Real-world Generalization

We conduct comprehensive experiments on the FMB dataset under various composite degradation scenarios. Notably, these test data are collected from real-world environments and represent degradation patterns not encountered in the training set. During evaluation, we rely exclusively on visual prompts automatically extracted from the input images, demonstrating our method's ability to handle unseen degradation conditions without manual intervention, as shown in Fig. 10.

## A.5 Effect of Severity Level Granularity

To ensure our model learns to discern fine-grained degradation severity, we train it on a dense set of four anchor degradation levels (1&4&7&10) from the DDL-12 dataset. This design is supported by a comparative analysis against models trained on sparser subsets of only two levels, specifically the moderate pair (4&7) or the extreme pair (1&10). As shown in Tab. 7, these sparser training regimes lead to a notable drop in generalization performance on real-world scenarios, particularly when only the extreme levels (1&10) are used. These results indicate the importance of dense sampling of training levels for robust generalization.

Table 7: Comparison of models trained with varying degradation levels on real-world datasets.

| | LLVIP | | | | FMB | | | |
|---|---|---|---|---|---|---|---|---|
| Level numbers | CLIP-IQA | MUSIQ | TReS | SD | CLIP-IQA | MUSIQ | TReS | SD |
| 2 (1&10) | 0.320 | 53.498 | 61.889 | 54.665 | 0.192 | 52.839 | 63.115 | 50.122 |
| 2 (4&7) | 0.332 | 55.364 | 64.382 | 55.742 | 0.207 | 52.771 | 63.328 | 50.429 |
| 4 (1&4&7&10) | **0.347** | **55.890** | **65.014** | **56.657** | **0.208** | **53.034** | **63.704** | **51.707** |

## A.6 Object Detection

As illustrated in Fig. 11, our fusion method achieves superior object detection performance with consistently high confidence scores. Notably, the detector successfully identifies all pedestrians in challenging scenarios, including heavily occluded cases where other methods fail. These results

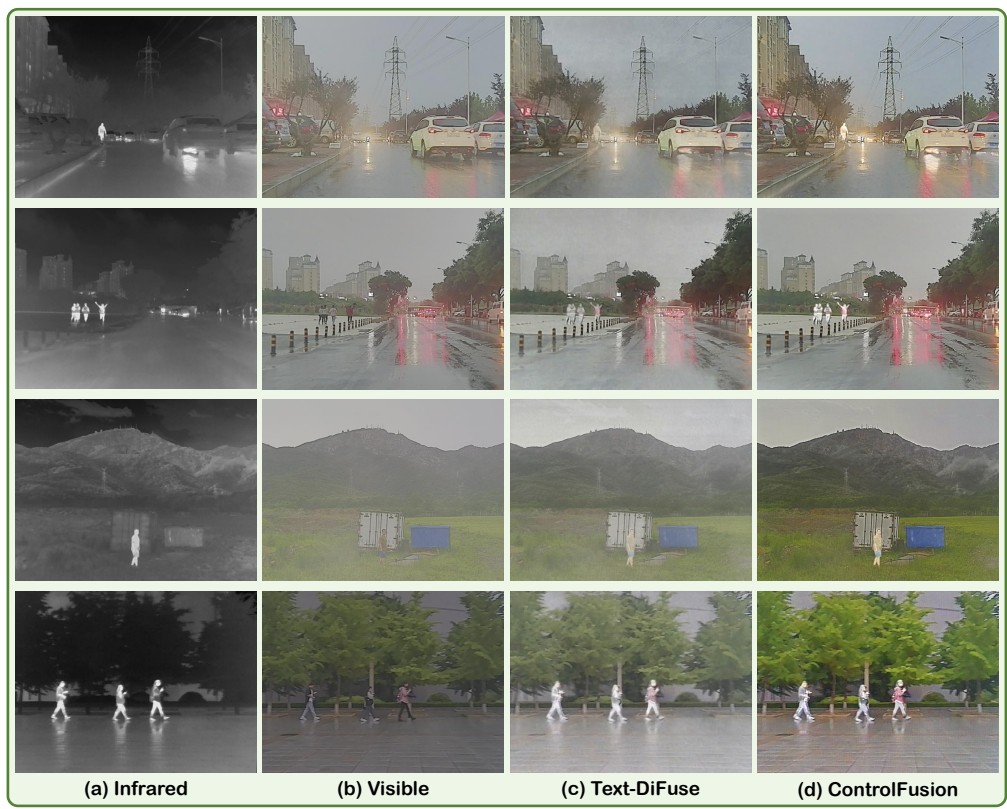

Figure 10: Real-world composite degradation scenario testing.

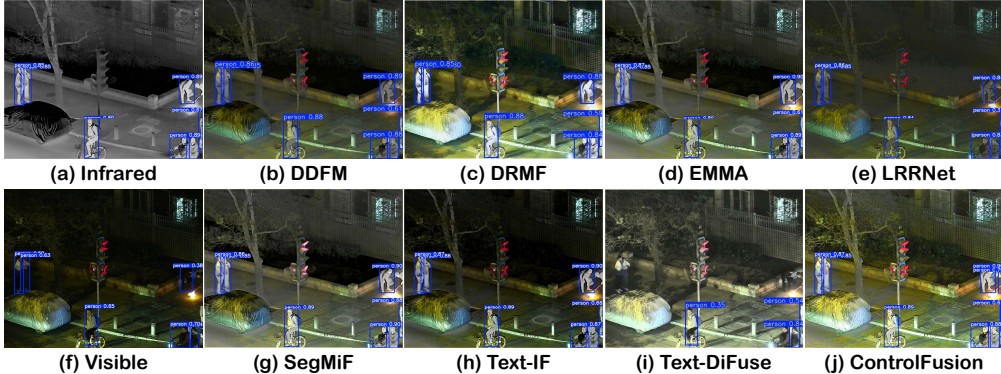

Figure 11: Visualization results of target detection.

quantitatively validate the effectiveness of our approach in preserving and enhancing critical visual information through the fusion process, particularly for small and obscured targets. The improved detection performance further confirms that our method maintains both thermal signatures and texture details essential for downstream vision tasks.

## A.7 Model Complexity Analysis

As demonstrated in Fig. 7 and Tab. 6, both CLIP and SFVA (*i.e.*, visual–text prompt alignment) yield substantial improvements in restoration and fusion performance. We analyze the computational cost of these components in Tab. 8. Although CLIP accounts for a significant number of parameters (102M, 55.9% of the total), they are frozen during training and thus introduce no training overhead. In terms of inference cost, CLIP and SFVA are responsible for a modest 0.81% and 7.5% of the total

Table 8: Overhead of individual components. Percentages indicate the relative share in the full model.

| Components | Parm. (M) | FLOPs (G) | Time (s) |
|---|---|---|---|
| **CLIP-L/14@336px** | 102.00 (55.88%) | 13.088 (0.81%) | 0.0120 (6.55%) |
| **SFVA** | 15.27 (8.37%) | 122.32 (7.54%) | 0.0067 (3.66%) |
| **Full Model** | 182.54 | 1622.00 | 0.1833 |

FLOPs, respectively, and their combined runtime constitutes a small fraction of the total. Overall, both components offer a compelling trade-off, delivering notable performance gains for a moderate computational expense.

