# OpenReview forum: "ControlFusion: A Controllable Image Fusion Network with Language-Vision Degradation Prompts"
_NeurIPS.cc/2025/Conference — NeurIPS 2025 oral_

### Official Review · Reviewer_SAs2 · 2025-06-15

**Clarity:** 2
**Significance:** 2
**Originality:** 2
**Rating:** 5
**Confidence:** 3

**Summary:**

This paper introduces ControlFusion, a novel and controllable framework for infrared-visible image fusion (IVIF) that simultaneously performs image restoration and fusion. The core problem is the poor performance of existing methods on real-world images suffering from complex, composite degradations (e.g., noise, low-light, haze, blur) and their lack of flexibility to accommodate user-specific needs. The authors presents a physics-driven degradation model to construct a realistic training dataset (DDL-12), and then use it for training controllable image fusion network. The training stage involves textual-visual prompt alignment and prompt-modulated restoration & fusion to achieve the controllable image fusion with prompts.

**Questions:**

Please see the weaknesses.

**Ethical Concerns:**

["NO or VERY MINOR ethics concerns only"]

**Final Justification:**

Thank you to the authors for their efforts and detailed responses. The responses address most of my concerns. After reviewing the clarifications and peer reviewers' comments, I believe the paper meets the acceptance criteria. Therefore, I’ve decided to Accept.

**Limitations:**

Yes

**Quality:**

2

**Strengths And Weaknesses:**

Strengths

1. The paper addresses a critical and practical limitation of current image fusion techniques: the inability to handle complex, real-world degradations and the lack of user control. The core idea of using language-vision prompts to control the fusion process is interesting and effectively implemented.

2. The paper proposes a realistic imaging model based on physical principles like the Retinex theory and atmospheric scattering to synthesize a high-quality training dataset (DDL-12) that bridges the domain gap between synthetic and real-world data.

3. Extensive experiments demonstrate that ControlFusion significantly outperforms state-of-the-art methods in both fusion quality and degradation handling, while also providing user-driven customizability.


Weaknesses

1. The introduction of multiple sophisticated modules for CLIP, SFVA, and PMM inevitably increases the model's parameter count and computational overhead compared to simpler fusion networks. This could be a potential drawback for deployment on resource-constrained edge devices where real-time performance is critical. This naturally leads to a question about the performance-efficiency trade-off. While the performance improvement is significant, it is important to consider whether this improvement mainly comes from using more computatioal cost.

2. The paper's SFVA extracts features from two fundamentally different domains: spatial and frequency. The method to combine them is a simple concatenation followed by a linear layer. This leads to a question about its effectiveness, as spatial features (capturing local textures) and frequency features (representing global periodic patterns) have vastly different properties. A simple linear fusion might not be powerful enough to model their different characteristics. A comprehensive study on this (e.g., more sophisticated fusion mechanisms) may be necessary to strengthen the paper's contribution.

3. While the DDL-12 dataset is a synthetic dataset, the model trained with it may not be generalizable to real-world scenarios. A "domain gap" inevitably exists between these simulated degradations and the unpredictable and complex degradations found in real-captured images. To fully validate the model's practical utility and its superiority in bridging the domain gap, it would be beneficial to include more experiments on realistic benchmarks and real-cpatured images.

---

> ### Author Rebuttal · Authors · 2025-07-30
>
> # Q1: Computational Efficiency
> > Thanks for the thoughtful comment. In response, we compared the computational cost of different fusion methods with pre-enhancement in **Table r1**, including parameters, FLOPs, and average runtime. Although **our method is not the most efficient, it achieves acceptable inference efficiency, especially when compared with joint restoration-fusion methods such as Text-DiFuse, DRMF, and Text-IF.**
>
> > Furthermore, **Table r2** reports the computational overhead of individual components. Since we use the pretrained CLIP model, its parameters do not incur additional training costs, and its FLOPs and runtime remain within acceptable bounds. To reduce complexity, our model also provides a lightweight SFVA as the prompt branch, significantly reducing both parameters and runtime. In addition, the PMM module consists of only a few MLP layers and simple modulation operations, incurring negligible overhead, particularly in terms of FLOPs and runtime.
>
> > In summary, **our method improves fusion performance without imposing a significant computational burden**. The **computational overhead introduced by the core components**, including CLIP, SFVA, and PMM, in terms of FLOPs and runtime, **remains within a practically acceptable range**.
>
> **Table r1.** Computational efficiency comparison with pre-enhancement. For fusion methods without restoration, we add the average cost of Instruct-IR and OneRestore. **Bold**: best; *Italic*: second-best.
> | |Methods|Parm.(M)|Flops(G)|Time(s)|
> |:-:|:-:|:-:|:-:|:-:|
> |**Restoration**|Instruct-IR|16.0|6.7|0.017|
> ||OneRestore|6.0|1.3|0.004|
> |**Fusion**|DDFM|563.7|5224|34.51|
> ||DRMF|170.9|4575|1.979|
> ||EMMA|*12.5*|*45.5*|**0.048**|
> ||LRRNet|**11.0**|**18.2**|*0.096*|
> ||SegMiF|56.0|358|0.158|
> ||Text-IF|89.0|302|0.157|
> ||Text-DiFuse|157.4|4872|31.63|
> ||Ours|182.5|348|0.183|
>
> **Table r2.** Computational overhead of individual components. Values in parentheses indicate the percentage relative to the full model.
> |Components|Parm.(M)|Flops(G)|Time(s)|
> |:-:|:-:|:-:|:-:|
> |CLIP-L/14@336px|102.00 (55.88%)|13.088 (**3.77%**)|0.0120 (**6.55%**)|
> |SFVA|15.27 (**8.37%**)|26.095 (**7.51%**)|0.0067 (**3.66%**)|
> |PMM|6.94 (**3.80%**)|0.014 (**0.004%**)|0.0001 (**0.05%**)|
> |Full Model|182.54|347.57|0.1833|
>
> # Q2: Spatial-Frequency Fusion Mechanisms
> > Thank you for your insightful comment regarding the fusion mechanism adopted in our SFVA module for integrating spatial and frequency features. To address this concern, we conducted additional experiments by introducing two more sophisticated fusion mechanisms: **1) Weighted Fusion:** We first enhance the extracted spatial and frequency features using ResBlocks, and then learn adaptive fusion weights for feature aggregation. **2) Attention Fusion:** Inspired by SwinFusion [r1], we incorporate a cross-attention mechanism to facilitate more effective interaction and better integration between spatial and frequency features.
>
> > We first evaluated these fusion mechanisms by comparing the cosine similarity between the visual prompts and their corresponding textual prompts across various degradation types and severity levels. The results, shown in **Tables r3** and **r4**, indicate that the more sophisticated fusion strategies lead to better alignment between visual and textual prompts, suggesting improved degradation characterization.
>
> > It is worth noting that, as shown in **Table r5**, these improvements come at the cost of **significantly increased computational complexity**. Both the weighted and attention fusion mechanisms **introduce more than 50% additional parameters, FLOPs, and runtime** compared to the simple concatenation strategy. However, **the actual gains in fusion performance brought  by these more complex mechanisms are marginal.**
>
> > Although advanced fusion mechanisms offer some benefits, our current design strikes a practical balance between effectiveness and computational efficiency. **We sincerely appreciate the reviewer’s suggestion and will explore lightweight yet effective spatial–frequency fusion strategies in future work.**
>
> **Table r3.** Cosine similarity between visual and textual prompts across various degradation types at medium severity across different fusion mechanisms.
> |Scenarios|Weighted Fusion|Attention Fusion|Ours|
> |:-:|:-:|:-:|:-:|
> |VI(Rain)|0.9137|**0.9172**|0.899|
> |VI(Haze)|0.8904|**0.8927**|0.8862|
> |VI(Low-light)|0.8991|**0.9005**|0.8939|
> |VI(Over-exposure, OE)|**0.9216**|0.918|0.9156|
> |VI(Random noise, RN)|**0.9387**|0.9362|0.9359|
> |IR(Stripe noise, SN)|**0.9302**|0.9293|0.9285|
> |IR(Low-contrast, LC)|**0.9511**|0.9507|0.9492|
> |VI(LL&RN)|0.9129|**0.9131**|0.9125|
> |VI(OE)&IR(LC)|**0.9184**|0.9123|0.9073|
> |VI(RH)&IR(RN)|**0.9351**|0.9275|0.9208|
>
> **Table r4.** Cosine similarity between visual and textual prompts under varying degradation levels in the low-light scenario across different fusion mechanisms.
> |Degradation Level|Weighted Fusion|Attention Fusion|Ours|
> |:-:|:-:|:-:|:-:|
> |Level 1|**0.909**|0.8951|0.8891|
> |Level 4|0.9296|**0.931**|0.9147|
> |Level 7|0.9416|**0.9458**|0.9356|
> |Level 10|0.9511|**0.9585**|0.9439|
>
> **Table r5.** Comparison of fusion performance and computational efficiency among different fusion mechanisms in SFVA.
> | |Weighted Fusion|Attention Fusion|Ours|
> |:-:|:-:|:-:|:-:|
> |**Computational Efficiency**| | | |
> |Parm.(M)|26.925 (+76.3%)|24.72 (+61.8%)|**15.274**|
> |Flops(G)|52.26 (+100.3%)|51.781 (+98.4%)|**26.095**|
> |Time(s)|0.0103 (+53.7%)|0.0098 (+46.3%)|**0.0067**|
> |**VI(LL&RN)**| | | |
> |CLIP-IQA|**0.227 (+0.89%)**|0.224|0.225|
> |MUSIQ|49.294|**49.403 (+0.14%)**|49.333|
> |TReS|**49.604 (+0.18%)**|49.595|49.513|
> |EN|7.132|**7.138 (+0.38%)**|7.111|
> |**VI(OE)&IR(LC)**| | | |
> |CLIP-IQA|0.186|0.186|**0.187**|
> |MUSIQ|**50.513(+0.07%)**|50.507|50.479|
> |TReS|50.327|**50.373 (+0.15%)**|50.298|
> |SD|50.873|50.938|**50.955**|
> |**VI(RH)&IR(RN)**| | | |
> |CLIP-IQA|0.179|**0.181 (+1.12%)**|0.179|
> |MUSIQ|50.121|**50.204 (+0.19%)**|50.107|
> |TReS|**51.129 (+0.07%)**|50.953|51.091|
> |SD|55.614|**55.71 (+0.53%)**|55.417|
>
> [r1] Ma J, Tang L, Fan F, et al. SwinFusion: Cross-domain long-range learning for general image fusion via swin transformer[J]. IEEE/CAA JAS, 2022, 9(7): 1200-1217.
>
> # Q3: Generalization to Real-world Scenarios
> > Thank you for the valuable suggestion. Although our model is trained solely on the synthetic DDL-12 dataset, it generalizes well to real-world scenarios, as evidenced by the qualitative results in **Figure 5** and **Figure 10 (Appendix)**.
>
> > To demonstrate the practical utility of our method and its effectiveness in bridging the domain gap, we conducted evaluations on real nighttime scenarios from the MSRS dataset and a real weather-degraded benchmark **AWMM-100k** [r2], as shown in **Tables r6** and **r7**, respectively. The MSRS dataset suffers from low-light and noise degradations, while AWMM-100k contains severe haze and snow conditions. As shown in the results, our method achieves clear advantages on these real-world benchmarks. This indicates that **the proposed synthetic degradation scheme, grounded in physical imaging models, effectively reduces the domain gap between simulated and real data, thereby improving the model's real-world generalization and applicability**.
>
> **Table r6.** Quantitative results on nighttime scenes from MSRS.
> |Methods|CLIP-IQA|MUSIQ|TReS|SD|
> |:-:|:-:|:-:|:-:|:-:|
> |DDFM|0.10|25.03|19.40|33.94|
> |DRMF|0.10|27.52|16.05|46.72|
> |EMMA|0.09|26.71|18.51|42.66|
> |LRRNet|0.10|27.04|16.91|42.69|
> |SegMiF|0.09|27.40|*21.08*|44.31|
> |Text-IF|0.10|28.16|20.64|*49.75*|
> |Text-DiFuse|*0.12*|**28.62**|18.97|43.73|
> |Ours|**0.14**|*28.46*|**21.61**|**52.93**|
>
> **Table r7.** Quantitative results on real-world weather degradations from AWMM-100k.
> |Methods|CLIP-IQA|MUSIQ|TReS|SD|
> |:-:|:-:|:-:|:-:|:-:|
> |DDFM|0.23|41.92|53.86|36.01|
> |DRMF|0.23|50.05|56.67|39.73|
> |EMMA|0.25|47.16|49.72|40.40|
> |LRRNet|0.19|46.75|50.71|37.19|
> |SegMiF|*0.26*|*50.71*|54.31|33.31|
> |Text-IF|0.21|45.63|51.75|32.09|
> |Text-DiFuse|0.23|46.90|*57.71*|*41.53*|
> |Ours|**0.28**|**57.51**|**60.30**|**44.24**|
>
> [r2] Li X, Liu W, Li X, et al. All-weather multi-modality image fusion: Unified framework and 100k benchmark. *arXiv preprint arXiv:2402.02090* (2024).

---

> > ### Comment · Reviewer_SAs2 · 2025-08-07
> > **Official Comment by Reviewer SAs2**
> >
> > Thank you to the authors for their efforts and detailed responses. The responses address most of my concerns. After reviewing the clarifications and peer reviewers' comments, I believe the paper meets the acceptance criteria. Therefore, I’ve decided to Accept.

---

### Official Review · Reviewer_iEbx · 2025-06-24

**Clarity:** 3
**Significance:** 3
**Originality:** 3
**Rating:** 5
**Confidence:** 5

**Summary:**

This paper proposes a controllable image fusion network guided by language-vision prompts, named ControlFusion, to address the problem of complex degradations in real-world scenarios. It integrates a physics-driven degradation model, a spatial-frequency collaborative visual adapter (SFVA), and a prompt-modulated restoration and fusion network. The model is capable of handling compound degradations and supports both language and visual prompts for further control. Extensive experiments on multiple datasets demonstrate the model's fusion quality and robustness under various degradation conditions.

**Questions:**

Although Figure 8 demonstrates the superiority of visual prompts over textual prompts, I would encourage the authors to further clarify:

1. Whether the visual prompt remains robust across different levels of degradation?
2. Whether the visual prompt is capable of handling compound degradations? Besides, please address the concerns in the Weakness part.

**Ethical Concerns:**

["NO or VERY MINOR ethics concerns only"]

**Final Justification:**

The response effectively addressed my concerns, and I have decided to maintain my accept score.

**Limitations:**

Yes.

**Paper Formatting Concerns:**

N / A

**Quality:**

3

**Strengths And Weaknesses:**

## Strengths

1. The paper introduces an imaging degradation model based on Retinex theory and atmospheric scattering, covering various degradation types for both infrared and visible modalities.
2. ControlFusion can control the fusion process through textual and visual prompts, enabling the network to flexibly adapt to different types and levels of degradation.
3. The method uses a spatial-frequency visual adapter (SFVA) to extract visual degradation prompts and align them with CLIP-based text embeddings, which is well-designed and addresses the limitations of purely text-driven methods.
4. The authors evaluate the model across multiple datasets (MSRS, LLVIP, RoadScene, FMB), multiple metrics (EN, SD, VIF, Qabf, CLIP-IQA, MUSIQ, TReS), and both with/without pre-enhancement. Ablation studies and real-world generalization are also included.

## Weaknesses

### Major Weaknesses

1. Although the visual prompt pathway alleviates dependence on textual input, the prompt-modulated module still hinges on CLIP-based representations. How does this ensure the effective extraction of degradation prompts from the source images?
2. The model includes multi-stage Transformers, visual-text prompt alignment, and SFVA, which increases inference time and model complexity. The author should compare the parameters and time complexity of the model.
3. The proposed method supports user-provided text prompts that include degradation severity. However, the paper does not provide sufficient details on the design of the prompts, such as the structure of text templates or how multiple degradations and their combinations are represented.
4. In the experimental evaluation related to degradations, the dataset used appears to contain a mixture of real and synthetic degradations. Additionally, qualitative results in Figure 5 alone is not sufficiently convincing to demonstrate the method’s effectiveness under real-world scenarios. To better validate effectiveness and robustness, the authors are encouraged to provide both quantitative and qualitative comparisons on datasets only with real degradations, such as night-time scenes from MSRS.

### Minor Weaknesses

1. The notation in Equation (9) needs to be corrected. Specifically, $Q_{vi}$ and $K_{ir}$ appear twice in the equation, which is confusing and likely a typographical error.
2. Some sentences in the paper have punctuation issues. For example, in Section 5.3, the first sentence: "removing composite degradations in real scenes.." contains an incorrect punctuation mark.

---

> ### Author Rebuttal · Authors · 2025-07-30
>
> # Q1: Effectiveness of Image-Based Degradation Prompts
> > Thank you for the insightful question. In our SFVA and prompt-modulated module, CLIP-based representations act as a bridge between visual and textual prompts, allowing our model to alleviate its dependence on textual input during inference. To ensure that the degradation prompts extracted from source images are effective, we design our SFVA from two perspectives. On the one hand, we incorporate a **frequency-aware branch to extract informative and discriminative cues in the frequency domain**. On the other hand, we introduce MSE loss and cosine similarity loss to **align the visual prompts with their corresponding textual counterparts** during training, thereby improving the consistency and reliability of the visual degradation representations.
>
> > As presented in **Figure 8 (Appendix)**, **using either visual or textual prompts leads to similarly consistent visual results, suggesting that visual degradation prompts from the source images are indeed effective**. To further validate this, we compute the cosine similarity between visual and textual prompts across different degradation types and levels, with results presented in **Tables r1** and **r2**. As shown in the results, our similarity scores are consistently higher than those of OneRestore in all scenarios except for low-light. This improvement primarily stems from the introduction of **our frequency-aware branch**, which enables better perception of degradation types. It is worth emphasizing that OneRestore’s visual prompt branch can only recognize a limited set of degradation types, such as rain, haze, and low-light, as well as their combinations, but lacks the ability to model degradation severity. In contrast, our method not only identifies the degradation types but also captures the corresponding degradation levels. Notably, when degradations become more severe, our degree-level matching becomes increasingly precise,  as shown in **Table r2**, which is consistent with human visual perception.
>
> **Table r1.** Cosine similarity between visual and textual prompts across various degradation types at medium severity.
> |Scenarios|Ours|OneRestore|
> |:-:|:-:|:-:|
> |VI (Rain)|0.899|0.8703|
> |VI (Haze)|0.8862|0.8749|
> |VI (Low-light, LL)|0.8939|0.9016|
> |VI (Over-exposure, OE)|0.9156|0.0289|
> |VI (Random noise, RN)|0.9359|0.0651|
> |IR (Stripe noise, SN)|0.9285|/|
> |IR (Low-contrast, LC)|0.9492|/|
> |VI (Rain&Haze, RH)|0.8516|0.8427|
> |VI (LL&RN)|0.9125|0.0072|
> |VI (OE)&IR(LC)|0.9073|/|
> |VI (RH)&IR(RN)|0.9208|/|
>
> **Table r2.** Cosine similarity between visual and textual prompts under varying degradation levels in the low-light scenario.
> |Degradation Levels|Level 1|Level 4|Level 7|Level 10|
> |:-:|:-:|:-:|:-:|:-:|
> |Similarity|0.8891|0.9147|0.9356|0.9439|
> # Q2: Model Complexity Analysis
> > Thank you for the valuable suggestion. We compare the computational complexity of various methods in **Table r3**, including **parameters, FLOPs, and runtime**. Although **our method is not the most efficient, it achieves acceptable inference efficiency, particularly when compared to joint restoration-fusion methods, such as DRMF, Text-DiFuse, and Text-IF.** Furthermore, **Table r4** presents the overhead introduced by individual components. We observe that **CLIP** and **SFVA**,  as key components for visual-text prompt alignment, significantly improve performance, as shown in **Table 4 (IV)**, where *“w/o PMM”* refers to the model without aligned visual-text prompts. Particularly, **this performance gain is attained with a relatively modest increase in computational overhead.**
>
> **Table r3.** Computational efficiency comparison with pre-enhancement. For fusion methods without restoration, we add the average cost of Instruct-IR and OneRestore.  **Bold**: best; *Italic*: second-best.
> | |Methods|Parm.(M)|Flops(G)|Time(s)|
> |:-:|:-:|:-:|:-:|:-:|
> |**Restoration**|Instruct-IR|16.0|6.7|0.017|
> ||OneRestore|6.0|1.3|0.004|
> |**Fusion**|DDFM|563.7|5224|34.51|
> ||DRMF|170.9|4575|0.842|
> ||EMMA|*12.5*|*45.5*|**0.048**|
> ||LRRNet|**11.0**|**18.2**|*0.096*|
> ||SegMiF|56.0|358|0.158|
> ||Text-IF|89.0|302|0.157|
> ||Text-DiFuse|157.4|4872|31.63|
> ||Ours|182.5|348|0.183|
>
> **Table r4.** Overhead of individual components.
> |Components|Parm.(M)|Flops(G)|Time(s)|
> |:-:|:-:|:-:|:-:|
> |CLIP-L/14@336px|102.00 (55.88%)|13.088 (**3.77%**)|0.0120 (**6.55%**)|
> |SFVA|15.27 (**8.37%**)|26.095 (**7.51%**)|0.0067 (**3.66%**)|
> |Full Model|182.54|347.57|0.1833|
>
> # Q3: Prompt Design Details
> > We appreciate the reviewer’s valuable suggestion regarding the design details of our textual prompts. Our prompts are designed to guide the fusion network by indicating the **modality** affected, the **degradation type**, and its **severity**.
>
> > A typical prompt template for a **single degradation** is structured as follows: "We are performing infrared and visible image fusion, where the `modality` image suffers from grade-`severity` `degradation type` problem."
>
> Here:
> - **Modality** includes `infrared` or `visible`.
> - **Severity** is an integer from 1 to 10.
> - **Degradation types** include: `rain`, `haze`, `low-light`, `overexposure`, `random noise`, `stripe noise`, `low contrast`, etc.
>
> For **composite degradations**, we adopt extended templates such as:
> > "We are performing infrared and visible image fusion. Please handle grade-`severity A` `degradation type A` problem in the `modality A` image, and grade-`severity B` `degradation type B` degradation  in the `modality B` image."
>
> or:
>
> > "We are performing infrared and visible image fusion. Please address level-`severity` `degradation type A` and `degradation type B` degradations in the `modality` image."
>
> These templates are flexibly composed to represent both intra- and inter-modal degradation combinations.
>
> An instantiation example is:
> > *"We are performing infrared and visible image fusion. Please handle the grade-6 low-light degradation in the visible image, and the grade-8 stripe noise interference in the infrared image."*
>
> > To cultivate a robust generalization capability, we curated a diverse set of over **100 unique textual formulations** for each task. This prompt rephrasing strategy augments linguistic variability and improves the model’s robustness to diverse prompt styles during inference.
>
> > We will add these illustrations of our prompt templates and examples in the camera-ready version.
>
> # Q4: Real Degradation Evaluation
> > Thank you for the valuable suggestions. To address your concerns, we conducted additional evaluations on real nighttime scenes from the MSRS dataset. Due to the system’s text-only submission constraint, we regret that qualitative results cannot be presented here. Quantitative results are shown in **Table r5**. Our method achieves the best performance on **CLIP-IQA**, **TRes**, and **SD** metrics, while slightly trailing **Text-DiFuse** on the **MUSIQ** metric.
>
> > Furthermore, we conducted quantitative comparisons on the real weather-degraded benchmark **AWMM-100k** [r1], with the results shown in **Table r6**. Our method outperforms others across all metrics under real-world weather degradation conditions. Additionally, the corresponding qualitative results also demonstrate the clear advantages of our approach.
>
> > These experimental results collectively validate the effectiveness and robustness of our method in real-world scenarios.
>
> **Table r5.** Quantitative results on nighttime scenes from MSRS.
> |Methods|CLIP-IQA|MUSIQ|TReS|SD|
> |:-:|:-:|:-:|:-:|:-:|
> |DDFM|0.10|25.03|19.40|33.94|
> |DRMF|0.10|27.52|16.05|46.72|
> |EMMA|0.09|26.71|18.51|42.66|
> |LRRNet|0.10|27.04|16.91|42.69|
> |SegMiF|0.09|27.40|*21.08*|44.31|
> |Text-IF|0.10|28.16|20.64|*49.75*|
> |Text-DiFuse|*0.12*|**28.62**|18.97|43.73|
> |Ours|**0.14**|*28.46*|**21.61**|**52.93**|
>
> **Table r6.** Quantitative results on real-world weather degradations from AWMM-100k.
> |Methods|CLIP-IQA|MUSIQ|TReS|SD|
> |:-:|:-:|:-:|:-:|:-:|
> |DDFM|0.23|41.92|53.86|36.01|
> |DRMF|0.23|50.05|56.67|39.73|
> |EMMA|0.25|47.16|49.72|40.40|
> |LRRNet|0.19|46.75|50.71|37.19|
> |SegMiF|*0.26*|*50.71*|54.31|33.31|
> |Text-IF|0.21|45.63|51.75|32.09|
> |Text-DiFuse|0.23|46.90|*57.71*|*41.53*|
> |Ours|**0.28**|**57.51**|**60.30**|**44.24**|
>
> [r1] Li, Xilai, et al. All-weather multi-modality image fusion: Unified framework and 100k benchmark. arXiv preprint arXiv:2402.02090, 2024.
>
> # Q5: Writing and Notation Clarity
> > Thank you for pointing out the notation and punctuation issues. We will carefully revise the manuscript and fix these issues in the camera-ready version.
>
> As the submission system supports only TeX format, we are unable to upload additional figures. We apologize for this limitation and will incorporate the above updates and results in the main paper or appendix.

---

> > ### Comment · Reviewer_iEbx · 2025-08-04
> >
> > Thanks for the response. It effectively addressed my concerns, and I have decided to maintain my accept score.

---

### Official Review · Reviewer_wXp7 · 2025-06-29

**Clarity:** 3
**Significance:** 3
**Originality:** 4
**Rating:** 5
**Confidence:** 5

**Summary:**

This paper proposes a controllable fusion network guided by language-vision prompts, called ControlFusion. The main contributions include two aspects. First, they construct a degraded imaging model based on physical mechanisms, to simulate data with composite degradation for learning. Second, a prompt-modulated restoration and fusion network is designed to dynamically enhances features according to degradation prompts. As a result, this work can adapt to varying degradation levels, showing promising robust fusion performance. They conduct extensive experiments to prove the advantages of ControlFusion.

**Questions:**

1) In Figure 2, does "composite degradation" refer to the simulated data? The logical relationship among (I), (II), and (III) is somewhat unclear.
2) In Figure 3, the authors provide the frequency spectrum of images containing various types of degradation; it would be better to additionally present the frequency spectrum of the images without degradation for comparison.
3) Does training the model without the proposed DDL-12 dataset degrade its performance in real-world degradation scenarios?

**Ethical Concerns:**

["NO or VERY MINOR ethics concerns only"]

**Final Justification:**

The authors have addressed most of my concerns. And I think this is a good paper. So I suggest to accept this paper.

**Limitations:**

yes

**Quality:**

4

**Strengths And Weaknesses:**

Strengths:
1) Datasets involving composite degradation are highly valuable for advancing research in multimodal image fusion. This work customizes simulated imaging based on physical models, which facilitates the batch generation of the required data.
2) This work uses text as a degradation prompt, which specifies not only the type but also the level of degradation. This provides a highly flexible interface for controlling the fusion process. It is also very meaningful for the deployment of multimodal image fusion models in real-world degraded environments.
3) The proposed spatial-frequency visual adapter is interesting, as it enables the perception of degradation distribution within source images based on their frequency characteristics, which is crucial for subsequent information restoration and fusion.
4) The conducted experiments are relatively comprehensive and effectively demonstrate the advantages and distinctive features of the proposed method.

Weaknesses:
1) One notable feature of the proposed method is its ability to specify the level of degradation. However, it is unclear how users are expected to have prior knowledge of the degradation severity. This could limit the practical utility of the interface. It would be helpful if the authors could provide guidance for determining appropriate degradation levels. In addition, the authors should provide a standardized paradigm for constructing textual prompts to facilitate the reproducibility of the proposed method.
2) The imaging model presented in Section 3 is based on existing physical models, but its core innovation should be further emphasized. It is recommended that this part be integrated into the methodology section to strengthen its connection to the overall contribution of the proposed method.
3) This work constructs degraded data based on existing datasets such as MSRS, RoadScene, and LLVIP, assuming that the original data are clean. However, if the original data are not clean, how is this issue addressed? Would the trained models be affected by the inherent noise or degradation in the base datasets?
4) In the “Comparison without Pre-enhancement” Section, the qualitative comparison on the four mentioned datasets is lacking.
5) In the “Comparison with Pre-enhancement” Section, it would be better to include full-reference fusion metrics, since relying solely on some perceptual metrics such as SD and EN cannot fully reflect fusion performance under degraded scenarios.
6) To more comprehensively demonstrate the real-world reliability of the proposed method, please include real composite degradation data, such as the nighttime scenes from the MSRS dataset.
7) Average running time is recommended to provide comparative evaluations of computational efficiency.

---

> ### Author Rebuttal · Authors · 2025-07-30
>
> # Q1: Prompt Construction Paradigm
> > Thank you for the insightful comments. Some key degradation severity levels are defined as follows:
> - **Level 1**: Barely perceptible degradation.
> - **Level 4**: Degradation begins to interfere with human scene understanding.
> - **Level 7**: Degradation severely hinders human perception of the scene.
> - **Level 10**: Most useful information is completely obscured.
>
> > Other levels allow users to interpolate between these anchor levels to describe intermediate degradations.
>
> > Moreover, when users are unsure about the degradation severity, our method supports a fully automatic mode: the **SFVA** can be used directly to perceive the degradation type and severity from inputs. This design balances **user-controllable flexibility** with **automated practicality**, ensuring both fine-grained control and real-world usability.
>
> > Our prompts are designed to guide the fusion network by indicating the **modality** affected, the **degradation type**, and its **severity**.
>
> > A typical prompt template for a **single degradation** is: "We are performing infrared and visible image fusion, where the `modality` image suffers from grade-`severity` `degradation type` problem."
>
> > For **composite degradations**, we adopt extended templates such as: "We are performing infrared and visible image fusion. Please handle grade-`severity A` `degradation type A` in the `modality A` image, and grade-`severity B` `degradation type B` in the `modality B` image."
>
> > Further details on **Prompt Construction Paradigm** are provided in our response to **Reviewer (iEbx) Q3**.
>
> # Q2: Imaging Model Contribution
> >We appreciate the reviewer's suggestion. Unlike existing imaging models that focus on isolated degradations, our proposed unified imaging model integrates three physical degradation paradigms and accounts for various modality characteristics. This enables accurate representation of complex, composite degradations and provides enriched data for model training, thereby helping to bridge the gap between synthetic data and real-world degradations. **We will further emphasize this key contribution and integrate the imaging model into the methodology section to better highlight its relevance to the overall framework.**
>
> # Q3: Training Data Construction
> >Thank you for the thoughtful comment. To address potential degradations in the base datasets, we use the high-quality counterparts provided by Text-IF [r1] as clean references. Thus, the training process is not affected by inherent noise or degradation in base datasets.
>
> [r1] Yi, Xunpeng, et al. Text-if: Leveraging semantic text guidance for degradation-aware and interactive image fusion. CVPR. 2024: 27026-27035.
>
> # Q4: Qualitative Comparison without Pre-enhancement
> >We regret the omission of qualitative comparisons. We have conducted qualitative assessments without pre-enhancement on the four mentioned datasets, where our method more effectively preserves salient objects from IR and retains rich texture details from VI, with these benefits also evident in qualitative comparisons with enhancement. To provide a more intuitive evaluation of fusion performance, we will include the qualitative comparisons in the Appendix.
>
> # Q5: Full-reference Metrics
> >Thank you for the valuable suggestion. We report full-reference metrics **MI**, **Qabf**, and **VIF** for various algorithms across three representative degraded scenarios in **Table r1**. Our method consistently outperforms others in most cases, further validating its effectiveness. We will include full-reference metrics for all degraded scenarios in the camera-ready version.
>
> **Table r1.** Comparison of full-reference metrics in various degraded scenarios. **Bold**: best; *Italic*: second-best.
> |Methods|VI(LL&Noise)|||VI(OE)&IR(LC)|||VI(RH)&IR(Noise)|||
> |:-:|:-:|:-:|:-:|:-:|:-:|:-:|:-:|:-:|:-:
> ||MI|Qabf|VIF|MI|Qabf|VIF|MI|Qabf|VIF|
> |DDFM|2.27|0.40|0.57|2.28|0.39|0.53|1.86|0.37|0.58|
> |DRMF|1.89|0.38|0.59|1.94|0.42|0.58|2.18|0.39|0.67|
> |EMMA|2.38|0.41|*0.70*|2.19|0.48|0.62|2.37|0.42|0.48|
> |LRRNet|1.95|0.51|0.62|2.41|0.47|0.61|2.20|0.50|0.56|
> |SegMiF|2.52|0.47|0.68|2.38|*0.52*|**0.73**|2.48|0.47|0.68|
> |Text-IF|*2.72*|*0.56*|0.61|*2.61*|0.50|0.68|2.52|0.53|0.66|
> |Text-DiFuse|2.50|0.50|0.66|2.48|*0.52*|0.69|**2.70**|*0.54*|*0.70*|
> |Ours|**2.75**|**0.59**|**0.73**|**2.70**|**0.58**|**0.73**|*2.66*|**0.58**|**0.71**|
>
> # Q6: Real Composite Degradation
> >Thanks for the insightful suggestion. As shown in **Table r2**, we provide comparative results on nighttime scenes from the MSRS dataset. Our method achieves the best performance on **CLIP-IQA**, **TRes**, and **SD** metrics, while slightly trailing Text-DiFuse on the MUSIQ metric.
>
> >Furthermore, we conducted quantitative comparisons on the real weather-degraded benchmark **AWMM-100k** [r2], with the results shown in **Table r3**. Our method outperforms others across all metrics under real-world weather degradation conditions.
>
> >The above results collectively demonstrate the real-world reliability of our method in handling real composite degradations.
>
> **Table r2.** Quantitative results on nighttime scenes from MSRS.
> |Methods|CLIP-IQA|MUSIQ|TReS|SD|
> |:-:|:-:|:-:|:-:|:-:|
> |DDFM|0.10|25.03|19.40|33.94|
> |DRMF|0.10|27.52|16.05|46.72|
> |EMMA|0.09|26.71|18.51|42.66|
> |LRRNet|0.10|27.04|16.91|42.69|
> |SegMiF|0.09|27.40|*21.08*|44.31|
> |Text-IF|0.10|28.16|20.64|*49.75*|
> |Text-DiFuse|*0.12*|**28.62**|18.97|43.73|
> |Ours|**0.14**|*28.46*|**21.61**|**52.93**|
>
> **Table r3.** Quantitative results on real-world weather degradations from AWMM-100k.
> |Methods|CLIP-IQA|MUSIQ|TReS|SD|
> |:-:|:-:|:-:|:-:|:-:|
> |DDFM|0.23|41.92|53.86|36.01|
> |DRMF|0.23|50.05|56.67|39.73|
> |EMMA|0.25|47.16|49.72|40.40|
> |LRRNet|0.19|46.75|50.71|37.19|
> |SegMiF|*0.26*|*50.71*|54.31|33.31|
> |Text-IF|0.21|45.63|51.75|32.09|
> |Text-DiFuse|0.23|46.90|*57.71*|*41.53*|
> |Ours|**0.28**|**57.51**|**60.30**|**44.24**|
>
> [r2] Li, Xilai, et al. All-weather multi-modality image fusion: Unified framework and 100k benchmark. arXiv preprint arXiv:2402.02090, 2024.
>
> # Q7: Computational Efficiency
> >Thanks for the valuable suggestion. **Table r4** reports the computational efficiency of various methods with enhancement, including parameters, FLOPs, and runtime. Although **our method is not the most efficient, it maintains acceptable inference efficiency, especially compared with joint restoration-fusion methods such as DRMF, Text-DiFuse, and Text-IF.**
>
> **Table r4.** Computational efficiency comparison with pre-enhancement. For fusion methods without restoration, we add the average cost of Instruct-IR and OneRestore.
> | |Methods|Parm.(M)|Flops(G)|Time(s)|
> |:-:|:-:|:-:|:-:|:-:|
> |**Restoration**|Instruct-IR|16.0|6.7|0.017|
> ||OneRestore|6.0|1.3|0.004|
> |**Fusion**|DDFM|563.7|5224|34.51|
> ||DRMF|170.9|4575|0.842|
> ||EMMA|*12.5*|*45.5*|**0.048**|
> ||LRRNet|**11.0**|**18.2**|*0.096*|
> ||SegMiF|56.0|358|0.158|
> ||Text-IF|89.0|302|0.157|
> ||Text-DiFuse|157.4|4872|31.63|
> ||Ours|182.5|348|0.183|
>
> # Q8: Diagram Clarity Issue
> >Thank you for the question. We believe the reviewer refers to **Figure 1**. *Composite degradation* does not exclusively indicate simulated data. For example, low-light degradation in VI is real. Importantly, it also refers to the cases where both IR & VI are degraded, and VI suffers from multi-type degradations simultaneously.  (I), (II), and (III) respectively highlight the limitations of existing methods and our advantages under **real-world scenarios**, **composite degradations**, and **varying severity levels**. We will further clarify the definition and relationships in the camera-ready version.
>
> # Q9: Frequency Spectrum Comparison
> >Thank you for the valuable suggestion. We will add the frequency spectrum of clean images in the camera-ready version to better illustrate differences between various degradations.
>
> # Q10: Training Data Dependency
> >Thank you for the insightful question. To investigate the necessity and impact of the proposed DDL-12 dataset, we retrained our model on the **Enhanced Multi-Spectral Various Scenarios (EMS) dataset** [r1], which contains sensor- and illumination-related degradations but lacks the full range of complex degradations modeled in DDL-12.
>
> >**Table r5** presents the quantitative results of the retrained model on real nighttime scenarios from the MSRS dataset and real-world weather degradations from AWMM-100k. On the MSRS dataset, the model trained on EMS shows slight performance drops in CLIP-IQA, MUSIQ, and TRes, and a more notable degradation in the SD metric. This can be attributed to the limited diversity in EMS, which weakens the model’s ability to perceive and adapt to compound degradations and different degradation levels. In addition, the fusion output tends to a uniform brightness distribution, thereby reducing contrast and structural diversity, which leads to a significant decrease in SD.
>
> >On AWMM-100k, the performance drops more significantly across all metrics. This is primarily because EMS does not provide any degradation patterns or priors related to adverse weather conditions, making it difficult for the model to generalize to scenarios involving haze or snow.
>
> >**These findings highlight the importance of DDL-12, which offers diverse and physically grounded synthetic degradations, enabling better learning of degradation-aware representations and significantly improving real-world generalization.**
>
> **Table r5.** Quantitative comparison under real-world degradation scenarios with different training datasets.
> |Real-world Scenarios|Training Datasets|CLIP-IQA|MUSIQ|TReS|SD|
> |:-:|:-:|:-:|:-:|:-:|:-:|
> |**MSRS(Nighttime)**|EMS|0.12|27.49|21.53|40.13|
> ||DDL-12(Ours)|0.14|28.46|21.61|52.93|
> |**AWMM-100k(Haze/Snow)**|EMS|0.23|49.16|56.85|33.52|
> ||DDL-12(Ours)|0.28|57.51|60.30|44.24|
>
> As the submission system supports only TeX format, we are unable to upload additional figures. We apologize for this limitation and will incorporate the above updates and results in the main paper or appendix.

---

> > ### Comment · Reviewer_wXp7 · 2025-08-05
> >
> > Thanks for your rebuttal! My questions have been addressed. I will reconsider my score based on your response and the reviews from other reviewers. Nice work!

---

### Official Review · Reviewer_JT4e · 2025-07-06

**Clarity:** 3
**Significance:** 3
**Originality:** 3
**Rating:** 4
**Confidence:** 4

**Summary:**

In this paper, the authors aim to fuse infrared and visible images under complex, multi-stage degradations. Specifically, these degradations stem from illumination, weather, and sensor effects. Based on this modeling process, they created a synthetic training dataset. The training also incorporates both degradation prompts and automatic degradation detection, so that the text prompt and the degraded image each provide cues about the degradation.

**Questions:**

- Have you evaluated how the number of degradation intensity levels used during training affects your model’s generalization on real‐world datasets (e.g., FMB, LLVIP)? For instance, if you change from 4 levels to 2, 3, or more, how does the performance vary?

- How are those pre-enhancements applied? Line 227 just mentions the application of Instruct-IR and OneRestore. In Table 2, do you use only one of them, or different fusion methods has different pre-enhancement method? Meanwhile, have the authors tried applying degradation-specific SOTA methods sequentially as pre-enhancement?

- How do the authors ensure that the training data domain covers the degradations present in real-world datasets? For example, the synthetic dataset simulates rain as streaks, but in reality, raindrops are much more common and challenging. Similarly, the haze model used here is homogeneous, yet varying the depth scale and hyperparameters can produce a wide range of haze effects. How can the proposed dataset and method handle such complex real-world data given the limited diversity in the training set?

- It seems that the real-world testing does not include weather degradations. Can the authors justify this omission?

- How can the infrared branch data provide insightful observations that can be implicitly or explicitly leveraged in this fusion task? It seems the authors simply enlarged the training dataset and helped the model learn everything without considering sensor-specific priors that could be exploited.

**Ethical Concerns:**

["NO or VERY MINOR ethics concerns only"]

**Final Justification:**

Even though the diversity of real-world testing is limited, the method demonstrates solid performance on the target task. Most of my concerns have been addressed by the authors through additional experiments.

Therefore, I will maintain my original score of weak accept, as I believe it reflects the quality and contribution of this paper.

**Limitations:**

yes

**Quality:**

3

**Strengths And Weaknesses:**

- Strength:
The consideration of multi-stage degradations appears relatively new for image fusion tasks, and the proposed method explicitly addresses this, improving robustness in real-world scenarios.

As reported, the proposed method is effective on both synthetic and real-world datasets.

- Weakness
The source of its performance on real-world datasets is questionable, given the training data's limited diversity. Additionally, the real-world tests do not include weather-related degradations.

Design-wise, the method primarily leverages data priors by constructing a multi-degradation paired dataset, yet it completely overlooks potential priors inherent in the infrared data format, making the overall solution less insightful.

---

> ### Author Rebuttal · Authors · 2025-07-30
>
> # Q1: Effect of Severity Level Number
> >Thanks for the valuable comment. As shown in **Table r1**, we compared models trained with all 4 anchor levels (1, 4, 7, 10) against those trained with only 2 levels (1 & 10 or 4 & 7). Reducing the number of degradation levels weakens the model’s generalization ability on real-world datasets, particularly when only extreme levels (1 & 10) are used. This limits the model’s capacity to accurately perceive fine-grained degradation severity. Due to limitations in time and dataset (DDL-12 has 4 levels: 1, 4, 7, and 10), we are currently unable to explore more settings as suggested (e.g., 3 or 5-6 levels), but will investigate them in the camera-ready.
>
> **Table r1.** Comparison of models trained with varying degradation levels in real-world datasets.
> ||FMB|||LLVIP||||
> |:-:|:-:|:-:|:-:|:-:|:-:|:-:|:-:|
> |Level numbers|2(1,10)|2(4,7)|4(1,4,7,10)|2(1,10)|2(4,7)|4(1,4,7,10)|
> |CLIP-IQA|0.192|0.207|0.208|0.32|0.332|0.347|
> |MUSIQ|52.84|52.77|53.03|53.5|55.36|55.89|
> |TReS|63.12|63.33|63.7|61.89|64.38|65.01|
> |SD|50.12|50.43|51.71|54.67|55.74|56.66|
> # Q2: Pre-enhancement Method
> >Thanks for the thoughtful question. In **Table 2**, **for each degradation type**, we apply the **same pre-enhancement method** for all fusion models without built-in enhancement modules. Specifically, **OneRestore** is used for weather-related degradations, while **Instruct-IR** is applied to illumination and sensor-related degradations. For IR stripe noise, we additionally employ **WDNN** [r1] as preprocessing, given the scarcity of effective stripe noise removal methods. We will clarify this in camera-ready.
>
> >As reported in **Table r2**, we further provide comparison results using SOTA degradation-specific pre-enhancement methods under various degradation scenarios. Specifically, DRMF, Text-IF, and Text-DiFuse adopt their own built-in enhancement modules in low-light conditions. For rain, haze, and rain&haze scenarios, we employ additional pre-enhancement algorithms, as these methods lack native support for such degradations. In particular, we use **HVI-CIDNet (CVPR'25)** [r2] for low-light enhancement, **NeRD-Rain (CVPR'24)** [r3] for de-raining, and **IPC-Dehaze (CVPR'25)** [r4] for dehazing. While these SOTA methods can improve fusion quality to some extent under specific degradation scenarios, our method consistently achieves superior overall performance. This is mainly due to our ability to **jointly exploit complementary information across modalities**, enabling more accurate scene representations and better approximation of high-quality imagery.
> >Furthermore, as shown in **Tables r2(d)** and **r2(e)**, we explore the impact of sequential processing orders under compound degradations. Results indicate that the processing order does affect fusion performance to some extent, especially in the rain&haze scenario, where addressing rain before haze yields moderate performance gains.
>
> **Table r2.** Quantitative results in various degraded scenarios with degradation-specific SOTA pre-enhancement methods. **Bold**: best; *Italic*: second-best.
> |Scenarios|Metrics|DDFM|DRMF|EMMA|LRRNet|SegMiF|Text-IF|Text-DiFuse|Ours|
> |:-:|:-:|:-:|:-:|:-:|:-:|:-:|:-:|:-:|:-:|
> |**(a) VI(LL)**|CLIP-IQA|0.16|0.14|0.17|0.16|*0.18*|0.16|**0.19**|*0.18*|
> |**HVI-CIDNet**|MUSIQ|40.24|41.43|40.81|42.14|44.81|41.10|*46.73*|**48.42**|
> |**(CVPR'25)**|TReS|42.42|37.95|46.11|35.19|47.89|49.17|*50.13*|**51.07**|
> ||SD|33.68|38.29|42.94|44.57|47.23|47.26|*49.88*|**53.79**|
> |**(b) VI(Rain)**|CLIP-IQA|**0.21**|0.18|0.15|0.19|0.19|0.17|0.18|*0.20*|
> |**NeRD-Rain**|MUSIQ|40.42|*49.07*|47.21|45.90|43.19|42.87|41.95|**52.32**|
> |**(CVPR'24)**|TReS|47.64|*49.18*|46.10|44.14|*50.64*|48.13|50.44|**52.47**|
> ||SD|38.20|*42.87*|45.18|47.37|45.10|*50.12*|48.06|**50.90**|
> |**(c) VI(Haze)**|CLIP-IQA|0.21|*0.24*|0.23|0.16|0.19|0.18|0.22|**0.26**|
> |**IPC-Dehaze**|MUSIQ|44.72|**47.75**|42.39|45.26|*46.50*|44.09|40.15|45.99|
> |**(CVPR'25)**|TReS|48.74|*48.88*|39.51|46.07|47.57|48.21|42.67|**49.71**|
> ||EN|7.01|*7.43*|*7.70*|7.18|6.98|7.61|7.29|**7.81**|
> |**(d) VI(Rain&Haze)**|CLIP-IQA|0.16|*0.18*|0.17|0.17|0.15|*0.18*|*0.18*|**0.19**|
> |**IPC-Dehaze→NeRD-Rain**|MUSIQ|44.84|46.79|42.18|50.01|48.49|51.79|*53.39*|**54.29**|
> ||TReS|45.74|44.71|47.89|49.03|47.44|51.24|*53.84*|**54.47**|
> ||EN|7.02|*6.71*|6.84|7.58|*7.64*|7.13|*7.59*|**7.89**|
> |**(e) VI(Rain&Haze)**|CLIP-IQA|0.16|*0.18*|0.17|0.17|0.16|*0.18*|*0.18*|**0.19**|
> |**NeRD-Rain→IPC-Dehaze**|MUSIQ|45.01|*46.95*|42.48|49.94|48.78|52.02|*53.02*|**54.29**|
> ||TReS|45.99|*45.13*|48.04|49.35|47.71|51.30|*53.73*|**54.47**|
> ||EN|6.99|*6.80*|6.92|7.52|*7.73*|7.20|7.62|**7.89**|
>
> [r1] Guan J, et al. Wavelet deep neural network for stripe noise removal[J]. IEEE Access, 2019, 7: 44544-44554.
>
> [r2] Yan Q, et al. Hvi: A new color space for low-light image enhancement[C]. CVPR. 2025:5678-5687.
>
> [r3] Chen X, et al. Bidirectional multi-scale implicit neural representations for image deraining[C]. CVPR. 2024: 25627-25636.
>
> [r4] Fu J, et al. Iterative Predictor-Critic Code Decoding for Real-World Image Dehazing[C]. CVPR. 2025: 12700-12709.
> # Q3: Real-world Datasets
> >Thanks for the insightful comment. In dataset construction, we carefully modeled degradations across sensor-related artifacts, illumination changes, and weather conditions to approximate real-world scenarios. Still, **some degradations (e.g., extreme weather), remain challenging to simulate accurately with our current imaging model, which is an open issue shared across existing methods.**
>
> >In Eq.(5), $D_{vi}^{w}=P_{w}(I_{vi})=I_{vi}\cdot t+A(1-t)+R$, we provide a **general formulation** for weather-related degradations, where $R$ denotes the rain component. It typically manifests as rain streaks or raindrops. While we implement rain streaks, the model is readily extensible to raindrops or other patterns, ensuring flexibility to simulate more complex weather effects.
>
> >We adopt a standard homogeneous haze model, a common practice in many methods [r5, r6], due to its physical soundness, controllability, and reproducibility. To enhance diversity, we simulate a range of haze effects by using various depth maps, varying transmission functions, and multiple parameter settings (e.g., $\beta\in[0.5,2.0],A\in[0.3,0.9]$). Despite its simplicity, this model exhibits promising generalization in real-world hazy scenes, as shown in **Figure 10 (Appendix)** and **Table r4**(AWMM-100k [r7]). We acknowledge its limitations, and will explore more complex variants, such as non-uniform haze (e.g., patch-wise variable $\beta$) or learning haze distributions from real foggy images.
>
> >Although DDL-12 covers only specific degradation types, it spans diverse **degradation patterns** from 3 complementary dimensions, i.e., sensor, illumination, and weather. This structured modeling provides our model with diverse and robust priors for generating high-quality fusion results. Besides, rather than tailoring to specific degradations in the training data, our model is designed to generalize beyond specific synthetic priors. For instance, although our dataset includes only rain and haze, the learned weather priors enable the model to handle unseen weather-related degradations such as snow. As shown in **Table r3**, our method achieves promising performance on snow scenarios from the AWMM-100k benchmark, demonstrating its robustness and generalization capability.
>
> **Table r3.** Quantitative results on snow scenarios from AWMM-100k.
> |Methods|CLIP-IQA|MUSIQ|TReS|SD|
> |:-:|:-:|:-:|:-:|:-:|
> |DDFM|0.27|43.45|48.16|23.61|
> |DRMF|*0.33*|*56.47*|*53.73*|32.41|
> |EMMA|0.32|48.99|38.39|30.73|
> |LRRNet|0.31|47.03|51.99|23.59|
> |SegMiF|0.29|50.31|44.98|28.98|
> |Text-IF|0.31|51.03|49.44|31.46|
> |Text-DiFuse|0.27|52.25|43.38|*36.80*|
> |Ours|**0.36**|**57.30**|**54.88**|**38.52**|
>
> [r5] Fattal R. Single image dehazing[J]. ACM TOG, 2008, 27(3):1-9.
>
> [r6] He K, et al. Single image haze removal using dark channel prior[J]. IEEE TPAMI, 2010, 33(12):2341-2353.
>
> [r7] Li X, et al. All-weather multi-modality image fusion: Unified framework and 100k benchmark. arXiv preprint arXiv:2402.02090 (2024).
>
> # Q4: Weather Degradations
> >Thank you for the question. As shown in **Figure 10 (Appendix)**, we provide results under haze conditions, a common weather-related degradation. Existing multi-modal datasets do not include rain scenarios, and due to exposure time limitations, our multimodal sensors cannot effectively capture real rain streaks. Therefore, we are currently unable to present results under real rain degradation. To further demonstrate our method's effectiveness under real-world weather degradations, we conducted experiments on **AWMM-100k** [r7], a multi-modal dataset with real-world adverse weather. As shown in **Table r4**, our method achieves clear advantages under real-world adverse weather conditions.
>
> **Table r4.** Quantitative results on real-world weather degradations from AWMM-100k.
> |Methods|CLIP-IQA|MUSIQ|TReS|SD|
> |:-:|:-:|:-:|:-:|:-:|
> |DDFM|0.23|41.92|53.86|36.01|
> |DRMF|0.23|50.05|56.67|39.73|
> |EMMA|0.25|47.16|49.72|40.40|
> |LRRNet|0.19|46.75|50.71|37.19|
> |SegMiF|*0.26*|*50.71*|54.31|33.31|
> |Text-IF|0.21|45.63|51.75|32.09|
> |Text-DiFuse|0.23|46.90|*57.71*|*41.53*|
> |Ours|**0.28**|**57.51**|**60.30**|**44.24**|
>
> # Q5: Sensor-Specific Priors
> >Thank you for your insightful comment. Our method does not solely rely on enlarging the training dataset but strategically leverages sensor-specific priors. First, we incorporate infrared-specific characteristics, such as stripe noise, when constructing degraded infrared data, enabling the model to effectively handle modality-specific degradations. Second, our fusion model implicitly exploits high-quality infrared properties, such as high contrast, to guide visible image restoration tasks like low-light enhancement and dehazing, where contrast enhancement is a key objective. These designs ensure that sensor-specific information is effectively utilized, beyond merely scaling the dataset.

---

> > ### Comment · Reviewer_JT4e · 2025-08-07
> >
> > Thanks for the authors' detailed rebuttal and explanation. The response addresses all of my current concerns, and I will update my rating accordingly.

---

### Decision · Program_Chairs · 2025-09-17

**Decision:**

Accept (oral)

**Comment:**

This work proposes a novel framework for image fusion designed to handle complex, multi-stage degradations in real-world scenarios. The paper's core contribution is a controllable fusion process guided by a language-vision prompt mechanism, which allows for automated degradation perception and fine-grained user control.

The reviewers unanimously acknowledged the novelty of the methodology and the significance of the problem being addressed. They initially raised several valid and critical concerns, primarily centered on the model's generalization from synthetic data to real-world conditions, its computational complexity, and the need for more rigorous validation. In response, the authors provided a comprehensive rebuttal. They made substantial efforts to address the reviewers' concerns, most notably by performing extensive experiments on challenging real-world benchmarks. These new results convincingly demonstrated the method's practical effectiveness and its ability to bridge the synthetic-to-real domain gap. lt is strongly recommended to incorporate the generalization experiments on real-world benchmarks and computational efficiency analysis into the camera-ready version.

The reviewers feel satisfied with the rebuttal, and the final scores are all positive: "Borderline accept", "Accept", "Accept", and "Accept". Therefore, the decision is to Accept this paper. Additionally, given its technical novelty, and its potential to influence research in areas such as multi-modal learning, image enhancement under complex degradations, and controllable image generation, the AC recommends this paper for an oral presentation.